# Adversarial Supervision Makes Layout-to-Image Diffusion Models Thrive

**Yumeng Li**[1,2]   **Margret Keuper**[2,3]   **Dan Zhang**[1,4]   **Anna Khoreva**[1]

[1]Bosch Center for Artificial Intelligence  [2]University of Mannheim
[3]Max Planck Institute for Informatics  [4]University of Tübingen
{yumeng.li, dan.zhang2, anna.khoreva}@de.bosch.com
 keuper@uni-mannheim.de
Project page: https://yumengli007.github.io/ALDM

## Abstract

Despite the recent advances in large-scale diffusion models, little progress has been made on the layout-to-image (L2I) synthesis task. Current L2I models either suffer from poor editability via text or weak alignment between the generated image and the input layout. This limits their usability in practice. To mitigate this, we propose to integrate **a**dversarial supervision into the conventional training pipeline of **L**2I **d**iffusion **m**odels (ALDM). Specifically, we employ a segmentation-based discriminator which provides explicit feedback to the diffusion generator on the pixel-level alignment between the denoised image and the input layout. To encourage consistent adherence to the input layout over the sampling steps, we further introduce the multistep unrolling strategy. Instead of looking at a single timestep, we unroll a few steps recursively to imitate the inference process, and ask the discriminator to assess the alignment of denoised images with the layout over a certain time window. Our experiments show that ALDM enables layout faithfulness of the generated images, while allowing broad editability via text prompts. Moreover, we showcase its usefulness for practical applications: by synthesizing target distribution samples via text control, we improve domain generalization of semantic segmentation models by a large margin ($\sim$12 mIoU points).

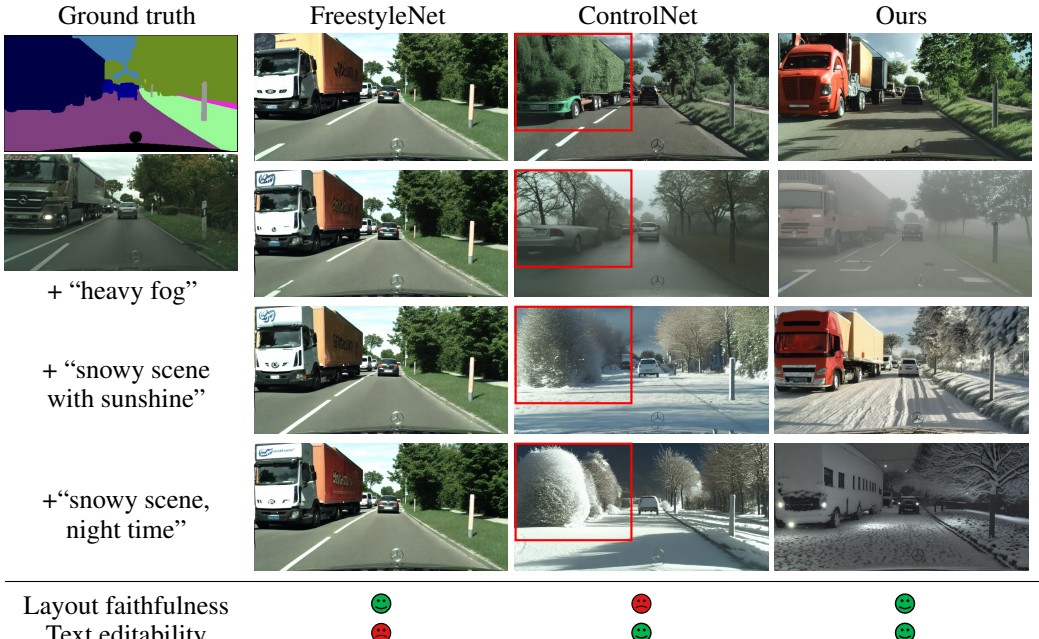

Figure 1: In contrast to prior L2I synthesis methods (Xue et al., 2023; Zhang & Agrawala, 2023), our **ALDM** model can synthesize faithful samples that are well aligned with the layout input, while preserving controllability via text prompt. Equipped with these both valuable properties, we can synthesize diverse samples of practical utility for downstream tasks, such as data augmentation for improving domain generalization of semantic segmentation models.

## 1 INTRODUCTION

Layout-to-image synthesis (L2I) is a challenging task that aims to generate images with per-pixel correspondence to the given semantic label maps. Yet, due to the tedious and costly pixel-level layout annotations of images, availability of large-scale labelled data for extensive training on this task is limited. Meanwhile, tremendous progress has been witnessed in the field of large-scale text-to-image (T2I) diffusion models (Ramesh et al., 2022; Balaji et al., 2022; Rombach et al., 2022). By virtue of joint vision-language training on billions of image-text pairs, such as LAION dataset (Schuhmann et al., 2022), these models have demonstrated remarkable capability of synthesizing photorealistic images via text prompts. A natural question is: can we adapt such pretrained diffusion models for the L2I task using a limited amount of labelled layout data while preserving their *text controllability* and *faithful alignment to the layout*? Effectively addressing this question will then foster the widespread utilization of L2I synthetic data.

Recently, increasing attention has been devoted to answer this question (Zhang & Agrawala, 2023; Mou et al., 2023; Xue et al., 2023). Despite the efforts, prior works have suffered to find a good trade-off between faithfulness to the layout condition and editability via text, which we also empirically observed in our experiments (see Fig. 1). When adopting powerful pretrained T2I diffusion models, e.g., Stable Diffusion (SD) (Rombach et al., 2022), for L2I tasks, fine-tuning the whole model fully as in (Xue et al., 2023) can lead to the loss of text controllability, as the large model easily overfits to the limited amount of training samples with layout annotations. Consequently, the model can only generate samples resembling the training set, thus negatively affecting its practical use for potential downstream tasks requiring diverse data. For example, for downstream models deployed in an open-world, variety in synthetic data augmentation is crucial, since annotated data can only partially capture the real environment and synthetic samples should complement real ones.

Conversely, when freezing the T2I model weights and introducing additional parameters to accommodate the layout information (Mou et al., 2023; Zhang & Agrawala, 2023), the L2I diffusion models naturally preserve text control of the pretrained model but do not reliably comply with the layout conditioning. In such case, the condition becomes a noisy annotation of the synthetic data, undermining its effectiveness for data augmentation. We hypothesize the poor alignment with the layout input can be attributed to the suboptimal MSE loss for the noise prediction, where the layout information is only implicitly utilized during the training process. The assumption is that the denoiser has the incentive to utilize the layout information as it poses prior knowledge of the original image and thus is beneficial for the denoising task. Yet, there is no direct mechanism in place to ensure the layout alignment. To address this issue, we propose to integrate **a**dversarial supervision on the layout alignment into the conventional training pipeline of **L**2I **d**iffusion **m**odels, which we name ALDM. Specifically, inspired by Sushko et al. (2022), we employ a semantic segmentation model based discriminator, explicitly leveraging the layout condition to provide a direct per-pixel feedback to the diffusion model generator on the adherence of the denoised images to the input layout.

Further, to encourage consistent compliance with the given layout over the sampling steps, we propose a novel multistep unrolling strategy. At inference time, the diffusion model needs to consecutively remove noise for multiple steps to produce the desired sample in the end. Hence, the model is required to maintain consistent adherence to the conditional layout over the sampling time horizon. Therefore, instead of applying discriminator supervision at a single timestep, we additionally unroll backward multiple steps over a certain time window to imitate the inference time sampling. This way the adversarial objective is designed over a time horizon and future steps are taken into consideration as well. Enabled by adversarial supervision over multiple sampling steps, our ALDM can effectively ensure consistent layout alignment, while maintaining initial properties of the text controllability of the large-scale pretrained diffusion model. We experimentally show the effectiveness of adversarial supervision for different adaptation strategies (Mou et al., 2023; Qiu et al., 2023; Zhang & Agrawala, 2023) of the SD model (Rombach et al., 2022) to the L2I task across different datasets, achieving the desired balance between layout faithfulness and text editability (see Table 1).

Finally, we demonstrate the utility of our method on the domain generalization task, where the semantic segmentation network is evaluated on unseen target domains, whose samples are sufficiently different from the trained source domain. By augmenting the source domain with synthetic images generated by ALDM using text prompts aligned with the target domain, we can significantly enhance the generalization performance of original downstream models, i.e., $\sim 12$ mIoU points on the Cityscapes-to-ACDC generalization task (see Table 4).

In summary, our main contributions include:

- We introduce adversarial supervision into the conventional diffusion model training, improving layout alignment without losing text controllability.
- We propose a novel multistep unrolling strategy for diffusion model training, encouraging better layout coherency during the synthesis process.
- We show the effectiveness of synthetic data augmentation achieved via ALDM. Benefiting from the notable layout faithfulness and text control, our ALDM improves the generalization performance of semantic segmenters by a large margin.

## 2 RELATED WORK

The task of layout-to-image synthesis (L2I), also known as semantic image synthesis (SIS), is to generate realistic and diverse images given the semantic label maps, which prior has been studied based on Generative Adversarial Networks (GANs) (Wang et al., 2018; Park et al., 2019; Wang et al., 2021; Tan et al., 2021; Sushko et al., 2022). The investigation can be mainly split into two groups: improving the conditional insertion in the generator (Park et al., 2019; Wang et al., 2021; Tan et al., 2021), or improving the discriminator's ability to provide more effective conditional supervision (Sushko et al., 2022). Notably, OASIS (Sushko et al., 2022) considerably improves the layout faithfulness by employing a segmentation-based discriminator. However, despite good layout alignment, the above GAN-based L2I models lack text control and the sample diversity heavily depends on the availability of expensive pixel-labelled data. With the increasing prevalence of diffusion models, particularly the large-scale pretrained text-to-image diffusion models (Nichol et al., 2022; Ramesh et al., 2022; Balaji et al., 2022; Rombach et al., 2022), more attention has been devoted to leveraging pretrained knowledge for the L2I task and using diffusion models. Our work falls into this field of study.

PITI (Wang et al., 2022) learns a conditional encoder to match the latent representation of GLIDE (Nichol et al., 2022) in the first stage and finetune jointly in the second stage, which unfortunately leads to the loss of text editability. Training diffusion models in the pixel space is extremely computationally expensive as well. With the emergence of latent diffusion models, i.e., Stable Diffusion (SD) (Rombach et al., 2022), recent works (Xue et al., 2023; Mou et al., 2023; Zhang & Agrawala, 2023) made initial attempts to insert layout conditioning into SD. FreestyleNet (Xue et al., 2023) proposed to rectify the cross-attention maps in SD based on the label maps, while it also requires fine-tuning the whole SD, which largely compromises the text controllability, as shown in Figs. 1 and 4. On the other hand, OFT partially updates SD, T2I-Adapter (Mou et al., 2023) and ControlNet (Zhang & Agrawala, 2023) keep SD frozen, combined with an additional adapter to accommodate the layout conditioning. Despite preserving the intriguing editability via text, they do not fully comply with the label map (see Fig. 1 and Table 1). We attribute this to the suboptimal diffusion model training objective, where the conditional layout information is only implicitly used without direct supervision. In light of this, we propose to incorporate the adversarial supervision to explicitly encourage alignment of images with the layout conditioning, and a multistep unrolling strategy during training to enhance conditional coherency across sampling steps.

Prior works (Xiao et al., 2022; Wang et al., 2023b) have also made links between GANs and diffusion models. Nevertheless, they primarily build upon GAN backbones, and the diffusion process is considered as an aid to smoothen the data distribution (Xiao et al., 2022), and stabilize the GAN training (Wang et al., 2023b), as GANs are known to suffer from training instability and mode collapse. By contrast, our ALDM aims at improving L2I diffusion models, where the discriminator supervision serves as a valuable learning signal for layout alignment.

## 3 ADVERSARIAL SUPERVISION FOR L2I DIFFUSION MODELS

L2I diffusion model aims to generate images based on the given layout. Its current training and inference procedure is inherited from unconditional diffusion models, where the design focus has been on how the layout as the condition is fed into the UNet for noise estimation, as illustrated in Fig. 2 (A). It is yet under-explored how to enforce the faithfulness of L2I image synthesis via direct loss supervision. Here, we propose novel adversarial supervision which is realized via 1) a semantic segmenter-based discriminator (Sec. 3.1 and Fig. 2 (B)); and 2) multistep unrolling of UNet (Sec. 3.2 and Fig. 2 (C)) to induce faithfulness already from early sampling steps and consistent adherence to the condition over consecutive steps.

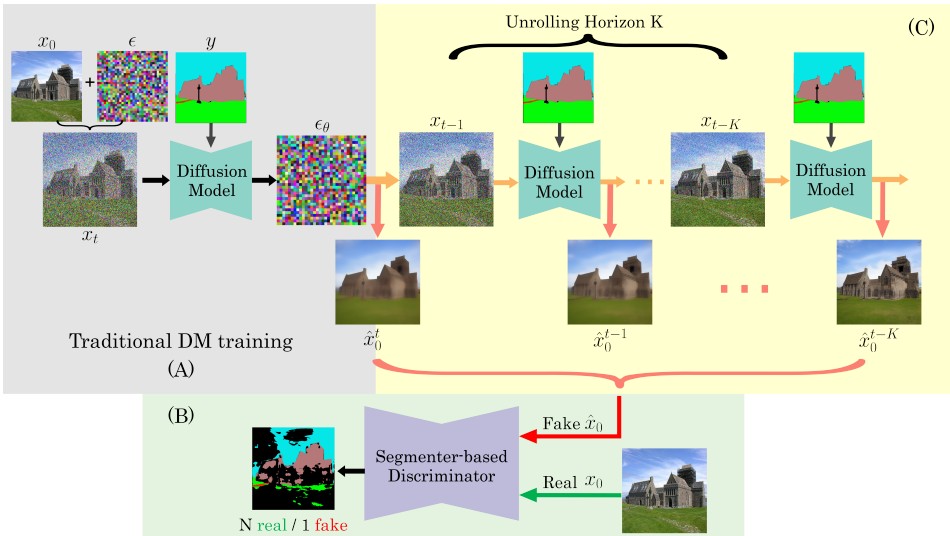

Figure 2: **Method overview.** To enforce faithfulness, we propose two novel training strategies to improve the traditional L2I diffusion model training (area (A)): adversarial supervision via a segmenter-based discriminator illustrated in area (B), and multistep unrolling strategy in area (C).

## 3.1 DISCRIMINATOR SUPERVISION ON LAYOUT ALIGNMENT

For training the L2I diffusion model, a Gaussian noise $\epsilon \sim N(0, I)$ is added to the clean variable $x_0$ with a randomly sampled timestep $t$, yielding $x_t$:

$$x_t = \sqrt{\alpha_t}x_0 + \sqrt{1 - \alpha_t}\epsilon, \tag{1}$$

where $\alpha_t$ defines the level of noise. A UNet (Ronneberger et al., 2015) denoiser $\epsilon_\theta$ is then trained to estimate the added noise via the MSE loss:

$$\mathcal{L}_{noise} = \mathbb{E}_{\epsilon \sim N(0,I),y,t}\left[\left\|\epsilon - \epsilon_\theta(x_t, y, t)\right\|^2\right] = \mathbb{E}_{\epsilon,x_0,y,t}\left[\left\|\epsilon - \epsilon_\theta(\sqrt{\alpha_t}x_0 + \sqrt{1 - \alpha_t}\epsilon, y)\right\|^2\right]. \tag{2}$$

Besides the noisy image $x_t$ and the time step $t$, the UNet additionally takes the layout input $y$. Since $y$ contains the layout information of $x_0$ which can simplify the noise estimation, it then influences implicitly the image synthesis via the denoising step. From $x_t$ and the noise prediction $\epsilon_\theta$, we can generate a denoised version of the clean image $\hat{x}_0^{(t)}$ as:

$$\hat{x}_0^{(t)} = \frac{x_t - \sqrt{1 - \alpha_t}\epsilon_\theta(x_t, y, t)}{\sqrt{\alpha_t}}. \tag{3}$$

However, due to the lack of explicit supervision on the layout information $y$ for minimizing $\mathcal{L}_{noise}$, the output $\hat{x}_0^{(t)}$ often lacks faithfulness to $y$, as shown in Fig. 3. It is particularly challenging when $y$ carries detailed information about the image, as the alignment with the layout condition needs to be fulfilled on each pixel. Thus, we seek direct supervision on $\hat{x}_0^{(t)}$ to enforce the layout alignment. A straightforward option would be to simply adopt a frozen pre-trained segmenter to provide guidance with respect to the label map. However, we observe that the diffusion model tends to learn a mean mode to meet the requirement of the segmenter, exhibiting little variation (see Table 3 and Fig. 6).

To encourage diversity in addition to alignment, we make the segmenter trainable along with the UNet training. Inspired by Sushko et al. (2022), we formulate an adversarial game between the UNet and the segmenter. Specifically, the segmenter acts as a discriminator that is trained to classify per-pixel class labels of real images, using the paired ground-truth label maps; while the fake images generated by UNet as in (Eq. (3)) are classified by it as one extra "fake" class, as illustrated in area (B) of Fig. 2. As the task of the discriminator is essentially to solve a multi-class semantic segmentation problem, its training objective is derived from the standard cross-entropy loss:

$$L_{Dis} = -\mathbb{E}\left[\sum_{c=1}^{N}\gamma_c \sum_{i,j}^{H \times W} y_{i,j,c} \log\left(Dis(x_0)_{i,jc}\right)\right] - \mathbb{E}\left[\sum_{i,j}^{H \times W} \log\left(Dis(\hat{x}_0^{(t)})_{i,j,c=N+1}\right)\right], \tag{4}$$

where $N$ is the number of real semantic classes, and $H \times W$ denotes spatial size of the input. The class-dependent weighting $\gamma_c$ is computed via inverting the per-pixel class frequency

$$\gamma_c = \frac{H \times W}{\sum \mathbb{E}\left[\mathbb{1}\left[y_{i,j,c} = 1\right]\right]}, \tag{5}$$

for balancing between frequent and rare classes. To fool such a segmenter-based discriminator, $\hat{x}_0^{(t)}$ produced by the UNet as in (Eq. (3)) shall comply with the input layout $y$ to minimize the loss

$$L_{adv} = -\mathbb{E}\left[\sum_{c=1}^{N} \gamma_c \sum_{i,j}^{H \times W} y_{i,j,c} \log\left(Dis(\hat{x}_0^{(t)})_{i,j,c}\right)\right]. \tag{6}$$

Such loss poses explicit supervision to the UNet for using the layout information, complementary to the original MSE loss. The total loss for training the UNet is thus

$$L_{DM} = L_{noise} + \lambda_{adv} L_{adv}, \tag{7}$$

where $\lambda_{adv}$ is the weighting factor. The whole adversarial training process is illustrated Fig. 2 (B). As the discriminator is improved along with UNet training, we no longer observe the mean mode collapsing as with the use of a frozen semantic segmenter. The high recall reported in Table 2 confirms the diversity of synthetic images produced by our method.

## 3.2 MULTISTEP UNROLLING

Admittedly, it is impossible for the UNet to produce high-quality image $\hat{x}_0^{(t)}$ via a single denoising step as in (Eq. (3)), especially if the input $x_t$ is very noisy (i.e., $t$ is large). On the other hand, adding such adversarial supervision only at low noise inputs (i.e., $t$ is small) is not very effective, as the alignment with the layout should be induced early enough during the sampling process. To improve the effectiveness of the adversarial supervision, we propose a multistep unrolling design for training the UNet. Extending from a single step denoising, we perform multiple denoising steps, which are recursively unrolled from the previous step:

$$x_{t-1} = \sqrt{\alpha_{t-1}}\left(\frac{x_t - \sqrt{1 - \alpha_t}\epsilon_\theta(x_t, y, t)}{\sqrt{\alpha_t}}\right) + \sqrt{1 - \alpha_{t-1}} \cdot \epsilon_\theta(x_t, y, t), \tag{8}$$

$$\hat{x}_0^{(t-1)} = \frac{x_{t-1} - \sqrt{1 - \alpha_{t-1}}\epsilon_\theta(x_{t-1}, y, t-1)}{\sqrt{\alpha_{t-1}}}. \tag{9}$$

As illustrated in area (C) of Fig. 2, we can repeat (Eq. (8)) and (Eq. (9)) $K$ times, yielding $\{\hat{x}_0^{(t)}, \hat{x}_0^{(t-1)}, ..., \hat{x}_0^{(t-K)}\}$. All these denoised images are fed into the segmenter-based discriminator as the "fake" examples:

$$L_{adv} = \frac{1}{K+1}\sum_{i=0}^{K} -\mathbb{E}\left[\sum_{c=1}^{N} \gamma_c y_c \log\left(Dis(\hat{x}_0^{(t-i)})_c\right)\right]. \tag{10}$$

By doing so, the denoising model is encouraged to follow the conditional label map consistently over the time horizon. It is important to note that while the number of unrolled steps $K$ is pre-specified, the starting time step $t$ is still randomly sampled.

Such unrolling process resembles the inference time denoising with a sliding window of size $K$. As pointed out by Fan & Lee (2023), diffusion models can be seen as control systems, where the denoising model essentially learns to mimic the ground-truth trajectory of moving from noisy image to clean image. In this regard, the proposed multistep unrolling strategy also resembles the advanced control algorithm - Model Predictive Control (MPC), where the objective function is defined in terms of both present and future system variables within a prediction horizon. Similarly, our multistep unrolling strategy takes future timesteps along with the current timestep into consideration, hence yielding a more comprehensive learning criteria.

While unrolling is a simple feed-forward pass, the challenge lies in the increased computational complexity during training. Apart from the increased training time due to multistep unrolling, the memory and computation cost for training the UNet can be also largely increased along with $K$.

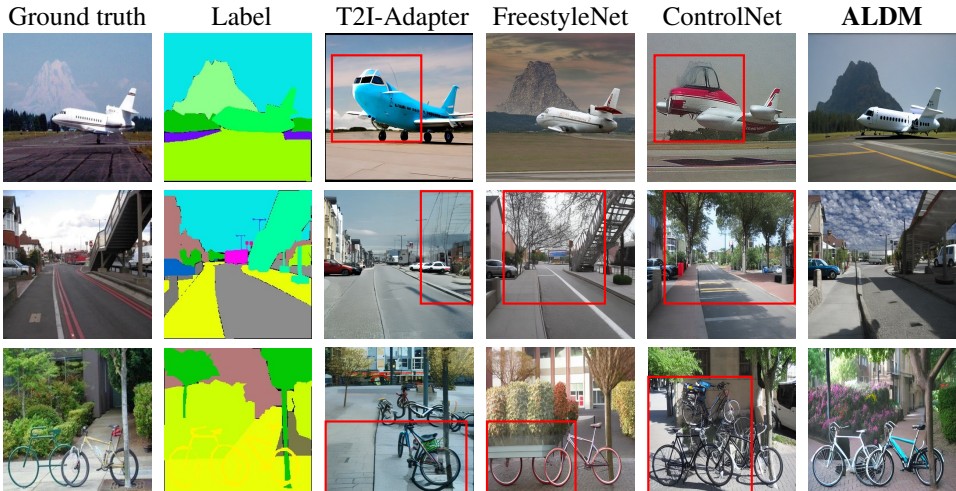

| Ground truth | Label | T2I-Adapter | FreestyleNet | ControlNet | **ALDM** |

Figure 3: Qualitative comparison of faithfulness to the layout condition on ADE20K.

Since the denoising UNet model is the same and reused for every step, we propose to simply accumulate and scale the gradients for updating the model over the time window, instead of storing gradients at every unrolling step. This mechanism permits to harvest the benefit of multistep unrolling with controllable increase of complexity during training.

**Implementation details.** We apply our method to the open-source text-to-image Stable Diffusion (SD) model (Rombach et al., 2022) so that the resulting model not only synthesizes high quality images based on the layout condition, but also accepts text prompts to change the content and style. As SD belongs to the family of latent diffusion models (LDMs), where the diffusion model is trained in the latent space of an autoencoder, the UNet denoises the corrupted latents which are further passed through the SD decoder for the final pixel space output, i.e., $\hat{x}_0 = \mathcal{D}(\hat{z}_0)$. We employ Uper-Net (Xiao et al., 2018) as the discriminator, nonetheless, we also ablate other types of backbones in Table 3. Since Stable Diffusion can already generate photorealistic images, a randomly initialized discriminator falls behind and cannot provide useful guidance immediately from scratch. We thus warm up the discriminator firstly, then start the joint adversarial training. In the unrolling strategy, we use $K = 9$ as the moving horizon. An ablation study on the choice of $K$ is provided in Table 5. Considering the computing overhead, we apply unrolling every 8 optimization steps.

## 4 EXPERIMENTS

Sec. 4.1 compares L2I diffusion models in terms of layout faithfulness and text editability. Sec. 4.2 further evaluates their use for data augmentation to improve domain generalization.

### 4.1 LAYOUT-TO-IMAGE SYNTHESIS

**Experimental Details.** We conducted experiments on two challenging datasets: ADE20K (Zhou et al., 2017) and Cityscapes (Cordts et al., 2016). ADE20K consists of 20K training and 2K validation images, with 150 semantic classes. Cityscapes has 19 classes, whereas there are only 2975 training and 500 validation images, which poses special challenge for avoiding overfitting and preserving prior knowledge of Stable Diffusion. Following ControlNet (Zhang & Agrawala, 2023), we use BLIP (Li et al., 2022b) to generate captions for both datasets.

By default, our ALDM adopts ControlNet (Zhang & Agrawala, 2023) architecture for layout conditioning. Nevertheless, the proposed adversarial training strategy can be combined with other L2I models as well, as shown in Table 1. For all experiments, we use DDIM sampler (Song et al., 2020) with 25 sampling steps. For more training details, we refer to Appendix A.1.

**Evaluation Metrics.** Following (Sushko et al., 2022; Xue et al., 2023), we evaluate the image-layout alignment via mean intersection-over-union (mIoU) with the aid of off-the-shelf segmentation networks. To measure the text-based editability, we use the recently proposed TIFA score (Hu et al., 2023), which is defined as the accuracy of a visual question answering (VQA) model, e.g.,

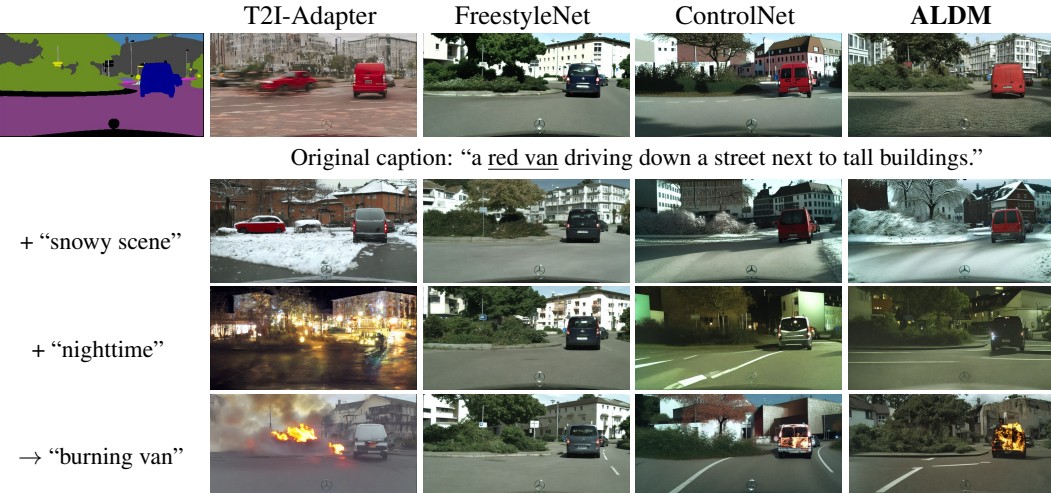

Figure 4: Visual comparison of text control between different L2I diffusion models on Cityscapes. Based on the image caption, we directly modify the underlined objects (indicated as →), or append a postfix to the caption (indicated as +). In contrast to prior work, ALDM can faithfully accomplish both global scene level modification (e.g., "snowy scene") and local editing (e.g., "burning van").

Table 1: Effect of adversarial supervision and multistep unrolling on different L2I synthesis adaptation methods. Best and second best are marked in bold and underline, respectively.

| Method | Cityscapes | | ADE20K | |
|---|---|---|---|---|
| | FID ↓ | mIoU↑ | FID↓ | mIoU↑ |
| OFT (Qiu et al., 2023) | 57.3 | 48.9 | **29.5** | 24.1 |
| + Adversarial supervision | 56.0 | 54.8 | 31.0 | 29.7 |
| + Multistep unrolling | 51.3 | 58.8 | 29.7 | 31.8 |
| T2I-Adapter (Mou et al., 2023) | 58.3 | 37.1 | 31.8 | 24.0 |
| + Adversarial supervision | 55.9 | 46.6 | 32.4 | 26.5 |
| + Multistep unrolling | 51.5 | 50.1 | 30.5 | 29.1 |
| ControlNet (Zhang & Agrawala, 2023) | 57.1 | 55.2 | 29.6 | 30.4 |
| + Adversarial supervision | **50.3** | 61.5 | 30.0 | 34.0 |
| + Multistep unrolling | 51.2 | **63.9** | 30.2 | **36.0** |

mPLUG (Li et al., 2022a), see Appendix A.2. Fréchet Inception Distance (FID) (Heusel et al., 2017), Precision and Recall (Sajjadi et al., 2018) are for assessing sample quality and diversity.

**Main Results.** In Table 1, we apply the proposed adversarial supervision and multistep unrolling strategy to different Stable Diffusion based L2I methods: OFT (Qiu et al., 2023), T2I-Adapter (Mou et al., 2023) and ControlNet (Zhang & Agrawala, 2023). Through adversarial supervision and multistep unrolling, the layout faithfulness is consistently improved across different L2I models, e.g., improving the mIoU of T2I-Adapter from 37.1 to 50.1 on Cityscapes. In many cases, the image quality is also enhanced, e.g., FID improves from 57.1 to 51.2 for ControlNet on Cityscapes. Overall, we observe that the proposed adversarial training complements different SD adaptation techniques and architecture improvements, noticeably boosting their performance. By default, ALDM represents ControlNet with adversarial supervision and multistep unrolling in other tables.

In Table 2, we quantitatively compare our ALDM with the other state-of-the-art L2I diffusion models: PITI (Wang et al., 2022), which does not support text control; and recent SD based FreestyleNet (Xue et al., 2023), T2I-Adapter and ControlNet, which support text control. FreestyleNet has shown good mIoU by trading off the editability, as it requires fine-tuning of the whole SD. Its poor editability, i.e., low TIFA score, is particularly notable on Cityscapes. As its training set is small and diversity is limited, FreestyleNet tends to overfit and forgets about the pretrained knowledge. This can be reflected from the low recall value in Table 2 as well. Both T2I-adapter and ControlNet freeze the SD, and T2I-Adapter introduces a much smaller adapter for the conditioning compared to ControlNet. Due to limited fine-tuning capacity, T2I-Adapter does not

Table 2: Quantitative comparison of the state-of-the-art L2I diffusion models. Best and second best are marked in bold and underline, respectively, while the worst result is in red. Our ALDM demonstrates competitive conditional alignment with notable text editability.

| Method | Cityscapes | | | | | ADE20K | | | | |
|---|---|---|---|---|---|---|---|---|---|---|
| | FID ↓ | mIoU↑ | P.↑ | R.↑ | TIFA↑ | FID↓ | mIoU↑ | P.↑ | R.↑ | TIFA↑ |
| PITI | n/a | n/a | n/a | n/a | ✗ | **27.9** | 29.4 | n/a | n/a | ✗ |
| FreestyleNet | 56.8 | **68.8** | **0.73** | 0.44 | 0.300 | 29.2 | **36.1** | 0.83 | 0.79 | 0.740 |
| T2I-Adapter | 58.3 | 37.1 | 0.55 | 0.59 | **0.902** | 31.8 | 24.0 | 0.79 | 0.81 | **0.892** |
| ControlNet | 57.1 | 55.2 | 0.61 | 0.60 | 0.822 | 29.6 | 30.4 | 0.84 | **0.84** | 0.838 |
| **ALDM (ours)** | **51.2** | 63.9 | 0.66 | **0.68** | 0.856 | 30.2 | 36.0 | **0.86** | 0.82 | 0.888 |

utilize the layout effectively, leading to low mIoU, yet it better preserves the editability, i.e., high TIFA score. By contrast, ControlNet improves mIoU while trading off the editability. In contrast, ALDM exhibits competitive mIoU while maintaining high TIFA score, which enables its usability for practical applications, e.g., data augmentation for domain generalization detailed in Sec. 4.2.

Qualitative comparison on the faithfulness to the label map is shown in Fig. 3. T2I-Adapter often ignores the layout condition (see the first row of Fig. 3), which can be reflected in low mIoU as well. FreestyleNet and ControlNet may hallucinate objects in the background. For instance, in the second row of Fig. 3, both methods synthesize trees where the ground-truth label map is sky. In the last row, ControlNet also generates more bicycles instead of the ground truth trees in the background. Contrarily, ALDM better complies with the layout in this case. Visual comparison on text editability is shown in Figs. 1 and 4. We observe that FreestyleNet only shows little variability and minor visual differences, as evidenced by the low TIFA score. T2I-Adapter and ControlNet on the other hand preserve better text control, nonetheless, they may not follow well the layout condition. In Fig. 1, ControlNet fails to generate the truck, especially when the prompt is modified. And in Fig. 4, the trees on the left are only sparsely synthesized. While ALDM produces samples that adhere better to both layout and text conditions, inline with the quantitative results.

Table 3: Ablation on the discriminator type.

| Method | Cityscapes | | ADE20K | |
|---|---|---|---|---|
| | FID↓ | mIoU↑ | FID↓ | mIoU↑ |
| ControlNet | 57.1 | 55.2 | 29.6 | 30.4 |
| + UperNet | 50.3 | 61.5 | 30.0 | 34.0 |
| + Segmenter | 52.9 | 59.2 | 29.8 | 34.1 |
| + Feature-based | 53.1 | 59.6 | 29.3 | 33.1 |
| + Frozen UperNet | - | - | 50.8 | 40.2 |

**Discriminator Ablation.** We conduct the ablation study on different discriminator designs, shown in Table 3. Both choices for the discriminator network: CNN-based segmentation network Uper-Net (Xiao et al., 2018) and transformer-based Segmenter (Strudel et al., 2021), improve faithfulness of the baseline ControlNet model. Instead of employing the discriminator in the pixel space, we also experiment with feature-space discriminator, which also works reasonably well. It has been shown that internal representation of SD, e.g., intermediate features and cross-attention maps, can be used for the semantic segmentation task (Zhao et al., 2023). We refer to Appendix A.3 for more details. Lastly, we employ a frozen semantic segmentation network to provide guidance directly. Note that this case is no longer adversarial training anymore, as the segmentation model does not update itself with the generator. Despite achieving high mIoU, the generator tends to learn a mean mode of the class and produce unrealistic samples (see Fig. 6), thus yielding high FID. In this case, the generator can more easily find a "cheating" way to fool the discriminator as it is not updating.

## 4.2 IMPROVED DOMAIN GENERALIZATION FOR SEMANTIC SEGMENTATION

We further investigate the utility of synthetic data generated by different L2I models for domain generalization (DG) in semantic segmentation. Namely, the downstream model is trained on a source domain, and its generalization performance is evaluated on unseen target domains. We experiment with both CNN-based segmentation model HRNet (Wang et al., 2020) and transformer-based Seg-Former (Xie et al., 2021). Quantitative evaluation is provided in Table 4, where all models except the oracle are trained on Cityscapes, and tested on both Cityscapes and the unseen ACDC. The oracle model is trained on both datasets. We observe that Hendrycks-Weather (Hendrycks & Dietterich, 2018), which simulates weather conditions in a synthetic manner, brings limited benefits. ISSA (Li et al., 2023b) resorts to simple image style mixing within the source domain. For models that accept

| Image | Ground truth | Baseline | Ours |

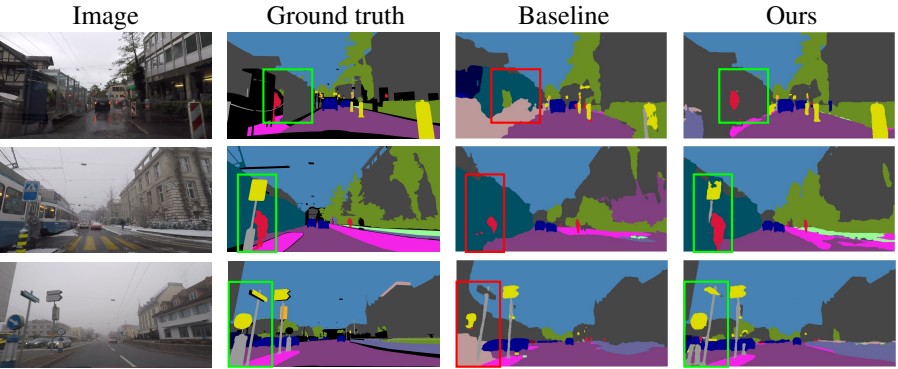

Figure 5: Semantic segmentation results of Cityscapes → ACDC generalization using HRNet. The HRNet is trained on Cityscapes only. Augmented with diverse synthetic data generated by our ALDM, the segmentation model can make more reliable predictions under diverse conditions.

Table 4: Comparison on domain generalization, i.e., from Cityscapes (train) to ACDC (unseen). mIoU is reported on Cityscapes (CS), individual scenarios of ACDC (Rain, Fog, Snow) and the whole ACDC. Hendrycks-Weather (Hendrycks & Dietterich, 2018) simulates weather conditions in a synthetic manner for data augmentation. Oracle model is trained on both Cityscapes and ACDC in a supervised manner, serving as an upper bound on ACDC (not Cityscapes) for the other methods. ALDM can consistently improve generalization performance of both HRNet and SegFormer.

| | HRNet (Wang et al., 2020) | | | | | SegFormer (Xie et al., 2021) | | | | |
| Method | CS | Rain | Fog | Snow | ACDC | CS | Rain | Fog | Snow | ACDC |
|---|---|---|---|---|---|---|---|---|---|---|
| Baseline (CS) | 70.47 | 44.15 | 58.68 | 44.20 | 41.48 | 67.90 | 50.22 | 60.52 | 48.86 | 47.04 |
| Hendrycks-Weather | 69.25 | 50.78 | 60.82 | 38.34 | 43.19 | 67.41 | 54.02 | 64.74 | 49.57 | 49.21 |
| ISSA | 70.30 | 50.62 | 66.09 | 53.30 | 50.05 | 67.52 | 55.91 | 67.46 | 53.19 | 52.45 |
| FreestyleNet | 71.73 | 51.78 | 67.43 | 53.75 | 50.27 | **69.70** | 52.70 | 68.95 | 54.27 | 52.20 |
| ControlNet | 71.54 | 50.07 | 68.76 | 52.94 | 51.31 | 68.85 | 55.98 | 68.14 | 54.68 | 53.16 |
| **ALDM (ours)** | **72.10** | **53.67** | **69.88** | **57.95** | **53.03** | 68.92 | **56.03** | **69.14** | **57.28** | **53.78** |
| Oracle (CS+ACDC) | 70.29 | 65.67 | 75.22 | 72.34 | 65.90 | 68.24 | 63.67 | 74.10 | 67.97 | 63.56 |

text prompts (FreestyleNet, ControlNet and ALDM), we can synthesize novel samples given the textual description of the target domain, as shown in Fig. 4. Nevertheless, the effectiveness of such data augmentation depends on the editability via text and faithfulness to the layout. FreestyleNet only achieves on-par performance with ISSA. We hypothesize that its poor text editability only provides synthetic data close to the training set with style jittering similar to ISSA's style mixing. While ControlNet allows text editability, the misalignment between the synthetic image and the input layout condition, unfortunately, can even hurt the performance. While mIoU averaged over classes is improved over the baseline, the per-class IoU shown in Table 7 indicates the undermined performance on small object classes, such as traffic light, rider and person. On those small objects, the alignment is noticeably more challenging to pursue than on classes with large area such as truck and bus. In contrast to it, ALDM, owing to its text editability and faithfulness to the layout, consistently improves across individual classes and ultimately achieves pronounced gains on mIoU across different target domains, e.g., 11.6% improvement for HRNet on ACDC. Qualitative visualization is illustrated in Fig. 5. The segmentation model empowered by ALDM can produce more reliable predictions under diverse weather conditions, e.g., improving predictions on objects such as traffic signs and person, which are safety critical cases.

## 5 CONCLUSION

In this work, we propose to incorporate adversarial supervision to improve the faithfulness to the layout condition for L2I diffusion models. We leverage a segmenter-based discriminator to explicitly utilize the layout label map and provide a strong learning signal. Further, we propose a novel multistep unrolling strategy to encourage conditional coherency across sampling steps. Our ALDM can well comply with the layout condition, meanwhile preserving the text controllability. Capitalizing these intriguing properties of ALDM, we synthesize novel samples via text control for data augmentation on the domain generalization task, resulting in a significant enhancement of the downstream model's generalization performance.

## ACKNOWLEDGEMENT

We would like to express our genuine appreciation to Shin-I Cheng for her dedicated support throughout the experimental testing.

## ETHICS STATEMENT

We have carefully read the ICLR 2024 code of ethics and confirm that we adhere to it. The method we propose in this paper allows to better steer the image generation during layout-to-image synthesis. Application-wise, it is conceived to improve the generalization ability of existing semantic segmentation methods. While it is fundamental research and could therefore also be used for data beyond street scenes (having in mind autonomous driving or driver assistance systems), we anticipate that improving the generalization ability of semantic segmentation methods on such data will benefit safety in future autonomous driving cars and driver assistance systems. Our models are trained and evaluated on publicly available, commonly used training data, so no further privacy concerns should arise.

## REPRODUCIBILITY STATEMENT

Regarding reproducibility, our implementation is based o publicly available models Rombach et al. (2022); Xiao et al. (2018); Zhang & Agrawala (2023); Song et al. (2020) and datasets Zhou et al. (2017); Cordts et al. (2016); Sakaridis et al. (2021) and common corruptions Hendrycks & Dietterich (2018). The implementation details are provided at the end of section 3, in the paragraph **Implementation Details**. Details on the experimental settings are given at the beginning of section 4. Further training details are given in the Appendix in section A. We plan to release the code upon acceptance.

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

# SUPPLEMENTARY MATERIAL

This supplementary material to the main paper is structured as follows:

- In Appendix A, we provide more experimental details for training and evaluation.

- In Appendix B, we include the ablation study on the unrolling step $K$, and more quantitative evaluation results.

- In Appendix C, we provide more visual results for both L2I task and improved domain generalization in semantic segmentation.

- In Appendix D, we discuss the failure cases of our approach, and potential solution for future research.

- In Appendix E, we discuss the theoretical connection with prior works and potential future research directions, which can be interesting for the community for further exploration and development grounded in our framework.

## A    EXPERIMENTAL DETAILS

### A.1    TRAINING DETAILS

We finetune Stable Diffusion v1.5 checkpoint and adopt ControlNet for the layout conditioning. All trainings are conducted on $512 \times 512$ resolution. For Cityscapes, we do random cropping and for ADE20K we directly resize the images. Nevertheless, we directly synthesize $512 \times 1024$ Cityscapes images for evaluation. We use AdamW optimizer and the learning rate of $1 \times 10^{-5}$ for the diffusion model, $1 \times 10^{-6}$ for the discriminator, and the batch size of $8$. The adversarial loss weighting factor $\lambda_{adv}$ is set to be $0.1$. The discriminator is firstly warmed up for 5K iterations on Cityscapes and 10K iterations on ADE20K. Afterward, we jointly train the diffusion model and discriminator in an adversarial manner. We conducted all training using 2 NVIDIA Tesla A100 GPUs.

### A.2    TIFA EVALUATION

Evaluation of the TIFA metric is based on the performance of the visual question answering (VQA) system, e.g. mPLUG (Li et al., 2022a). By definition, the TIFA score is essentially the VQA accuracy, given the question-answer pairs. To quantitatively evaluate the text editability, we design a list of prompt templates, e.g., appending "snowy scene" to the original image caption for image generation. Based on the known prompts, we design the question-answer pairs. For instance, we can ask the VQA model "What is the weather condition?", and compute TIFA score based on the accuracy of the answers.

### A.3    FEATURE-BASED DISCRIMINATOR FOR ADVERSARIAL SUPERVISION

Thanks to large-scale vision-language pretraining on massive datasets, Stable Diffusion (SD) (Rombach et al., 2022) has acquired rich representations, endowing it with the capability not only to generate high-quality images, but also to excel in various downstream tasks. Recent work VPD (Zhao et al., 2023) has unleashed the potential of SD, and leveraged its representation for visual perception tasks, e.g., semantic segmentation. More specifically, they extracted cross-attention maps and feature maps from SD at different resolutions and fed them to a lightweight decoder for the specific task. Despite the simplicity of the idea, it works fairly well, presumably due to the powerful knowledge of SD. In the ablation study, we adopt the segmentation model of VPD as the feature-based discriminator. Nevertheless, different from the joint training of SD and the task-specific decoder in the original VPD implementation, we only train the newly added decoder, while freezing SD to preserve the text controllability as ControlNet.

Table 5: Ablation on the unrolling step $K$. Overhead is measured as seconds per training iteration.

| | Cityscapes | | | ADE20K | | | |
| | FID↓ | mIoU↑ | TIFA↑ | FID↓ | mIoU↑ | TIFA↑ | Overhead |
|---|---|---|---|---|---|---|---|
| ControlNet | 57.1 | 55.2 | 0.822 | 29.6 | 30.4 | 0.838 | 0.00 |
| K = 0 | 50.3 | 61.5 | 0.894 | 30.0 | 34.0 | 0.904 | 0.00 |
| K = 3 | 54.9 | 62.7 | 0.856 | - | - | - | 1.55 |
| K = 6 | 51.6 | 64.1 | 0.832 | 30.3 | 34.5 | 0.898 | 3.11 |
| K = 9 | 51.2 | 63.9 | 0.856 | 30.2 | 36.0 | 0.888 | 4.65 |
| K = 15 | 50.7 | 64.1 | 0.882 | 30.2 | 36.9 | 0.825 | 7.75 |

Table 6: Quantitative comparison of different T2I diffusion models. P., R., and R.mIoU represent Precision, Recall and robust mIoU respectively.

| Method | FID ↓ | P.↑ | R.↑ | mIoU↑ | R.mIoU↑ | TIFA ↑ |
|---|---|---|---|---|---|---|
| FreestyleNet | 56.8 | 0.73 | 0.44 | 68.8 | 69.9 | 0.300 |
| T2I-Adapter | 58.3 | 0.55 | 0.59 | 37.1 | 44.7 | 0.902 |
| ControlNet | 57.1 | 0.61 | 0.60 | 55.2 | 57.3 | 0.822 |
| **ALDM** | 51.2 | 0.66 | 0.68 | 63.9 | 65.4 | 0.856 |

# B  MORE ABLATION AND EVALUATION RESULTS

## B.1  MULTISTEP UNROLLING ABLATION

For the unrolling strategy, we compare different number of unrolling steps in Table 5. We observe that more unrolling steps is beneficial for improving the faithfulness, as the model can consider more future steps to ensure alignment with the layout condition. However, the additional unrolling time overhead also increases linearly. Therefore, we choose $K = 9$ by default in all experiments.

## B.2  ABLATION ON FROZEN SEGMENTER

We ablate the usage of a *frozen* segmentation model, instead of joint training with the diffusion model. As quantitatively evaluated in Table 3, despite achieving good alignment with the layout condition, i.e., high mIoU, we observe that the diffusion model tends to lean a mean mode and produces unrealistic samples with limited diversity (see Fig. 6), thus yielding high FID values.

## B.3  ROBUST MIOU EVALUATION

Conventionally, the layout alignment evaluation is conducted with the aid of off-the-shelf segmentation networks trained on the specific dataset, thus may not be competent enough to make reliable predictions on more diverse data samples, as shown in Fig. 7. Therefore, we propose to employ a robust segmentation model trained with special data augmentation techniques (Li et al., 2023b), to more accurately measure the actual alignment.

We report the quantitative performance in Table 6. Notably, there is a large difference between the standard mIoU and robust mIoU, in particular for T2I-Adapter. From Fig. 7, we can see that T2I-Adapter occasionally generates more stylized samples, which do not comply with the Cityscapes style, and the standard segmenter has a sensitivity to this.

## B.4  COMPARISON WITH GAN-BASED L2I METHODS.

We additionally compare our method with prior GAN-based L2I methods in Table 8. It is worth-while to mention that all GAN-based approaches do not have text controllability, thus they can only produce samples resembling the training dataset, which constrains their utility on downstream tasks. On the other hand, our ALDM achieves the balanced performance between faithfulness to the layout condition and editability via text, rendering itself advantageous for the domain generalization tasks.

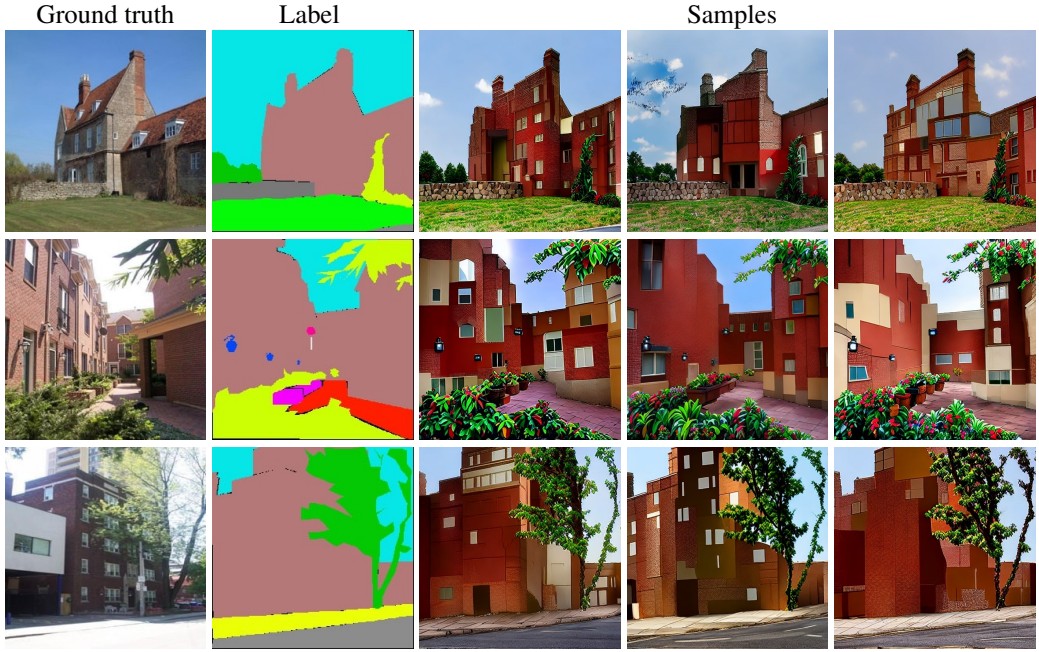

Figure 6: Visual results of using a *frozen* segmentation network, i.e., a pretrained UperNet (Xiao et al., 2018), to provide conditional guidance during diffusion model training. We can observe the mode collapse issue, where the diffusion model tends to learn to a mean mode and exhibits little variation in the generated samples.

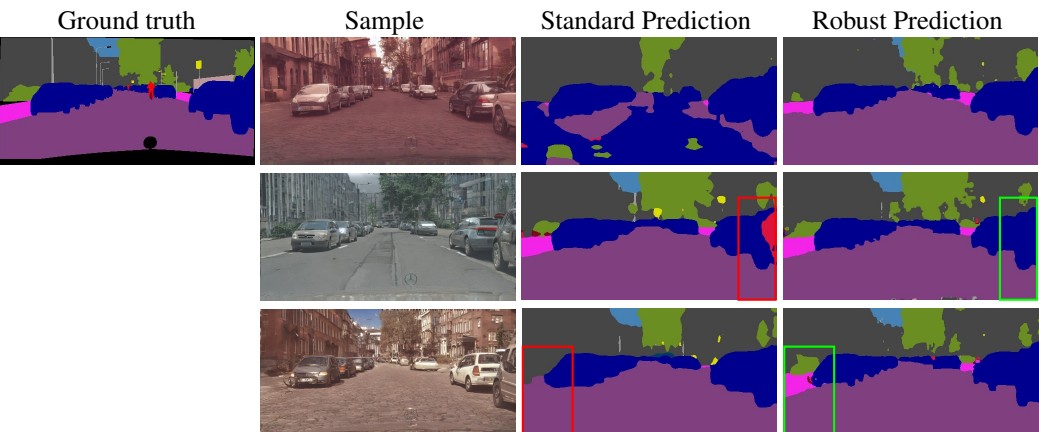

Figure 7: Comparison of standard segmenter and robust segmenter for layout alignment evaluation on synthesized samples of T2I-Adapter. When testing on more diverse images, the standard segmenter struggles to make reliable predictions, thus may lead to inaccurate evaluation.

## B.5 PER-CLASS IoU FOR SEMANTIC SEGMENTATION

In Table 7, we report per-class IoU of the object classes on Cityscapes. For ControlNet the misalignment between the synthetic image and the input layout condition, unfortunately, can hurt the segmentation performance on small and fine-grained object classes, such as bike, traffic light, traffic sign, rider, and person. While ALDM demonstrates better performance on those classes, which reflects that our method can better comply with the layout condition, as small and fine-grained object classes are typically more challenging in L2I task and pose higher requirements for the faithfulness to the layout condition.

Table 7: Per-class IoU of Cityscapes object classes. Numbers in red indicate worse IoU compared to the baseline. The best is marked in bold. Our ALDM has demonstrated better performance on small object classes, e.g., pole, traffic light, traffic sign, person, rider, which reflects our method can better comply with the layout condition, as small object classes are typically more challenging in L2I task and pose higher requirement for the faithfulness to the layout condition.

| Method | Pole | Traf. light | Traf. sign | Person | Rider | Car | Truck | Bus | Train | Motorbike | Bike |
|---|---|---|---|---|---|---|---|---|---|---|---|
| Baseline | 48.50 | 59.07 | 67.96 | 72.44 | 52.31 | 92.42 | 70.11 | 77.62 | 64.01 | 50.76 | 68.30 |
| ControlNet | 49.53 | 58.47 | 67.37 | 71.45 | 49.68 | 92.30 | **76.91** | **82.98** | 72.40 | 50.84 | 67.32 |
| **ALDM** | **51.21** | **60.50** | **69.56** | **73.82** | **53.01** | **92.57** | 76.61 | 81.37 | 66.49 | **52.79** | **68.61** |

Table 8: Quantitative comparison results with the state-of-the-art layout-to-image GANs and diffusion models (DMs). Our ALDM demonstrates competitive conditional alignment with notable text editability.

| | Method | Cityscapes | | | ADE20K | | |
|---|---|---|---|---|---|---|---|
| | | FID↓ | mIoU↑ | TIFA↑ | FID↓ | mIoU↑ | TIFA↑ |
| GANs | Pix2PixHD (Wang et al., 2018) | 95.0 | 63.0 | | 81.8 | 28.8 | |
| | SPADE (Park et al., 2019) | 71.8 | 61.2 | | 33.9 | 38.3 | |
| | OASIS (Schönfeld et al., 2020) | 47.7 | 69.3 | | 28.3 | 45.7 | |
| | SCGAN (Wang et al., 2021) | 49.5 | 55.9 | ✗ | 29.3 | 41.5 | ✗ |
| | CLADE (Tan et al., 2021) | 57.2 | 58.6 | | 35.4 | 23.9 | |
| | GroupDNet (Zhu et al., 2020) | 47.3 | 55.3 | | 41.7 | 27.6 | |
| DMs | PITI (Wang et al., 2022) | n/a | n/a | ✗ | 27.9 | 29.4 | ✗ |
| | FreestyleNet (Xue et al., 2023) | 56.8 | 68.8 | 0.300 | 29.2 | 36.1 | 0.740 |
| | T2I-Adapter (Mou et al., 2023) | 58.3 | 37.1 | 0.902 | 31.8 | 24.0 | 0.892 |
| | ControlNet (Zhang & Agrawala, 2023) | 57.1 | 55.2 | 0.822 | 29.6 | 30.4 | 0.838 |
| | **ALDM (ours)** | 51.2 | 63.9 | 0.856 | 30.2 | 36.0 | 0.888 |

# C   MORE VISUAL EXAMPLES

## C.1   LAYOUT-TO-IMAGE TASKS

In Fig. 8, we showcase more visual comparison on ADE20K across various scenes, i.e., outdoors and indoors. Our ALDM can consistently adhere to the layout condition.

Figure 9 presents visual examples of Cityscapes, which are synthesized via various textual descriptions with our ALDM, which can be further utilized on downstream tasks.

In Fig. 10, we demonstrate the editability via text of our ALDM. Our method enables both global editing (e.g., style or scene-level modification) and local editing (e.g., object attribute).

In Figs. 11 until 13, we provide qualitative comparison on the text editability between different L2I diffusion models on Cityscapes and ADE20K.

In Fig. 14, we compare our ALDM with GAN-based style transfer method ISSA (Li et al., 2023b). It can be observed that ALDM produces more realistic results with faithful local details, given the label map and text prompt. In contrast, style transfer methods require two images, and mix them on the global color style, while the local details, e.g., mud, and snow may not be faithfully transferred.

In Fig. 15, we provide visualization of the segmentation outputs from the discriminator. It can be seen that the discriminator can categorize some regions as "fake" class (in black), meanwhile it is also fooled by some areas, where the discriminator produces reasonable segmentation predictions matched with ground truth labels. Therefore, the discriminator can provide useful feedback such that the generator (diffusion model) produces realistic results meanwhile complying with the given label map.

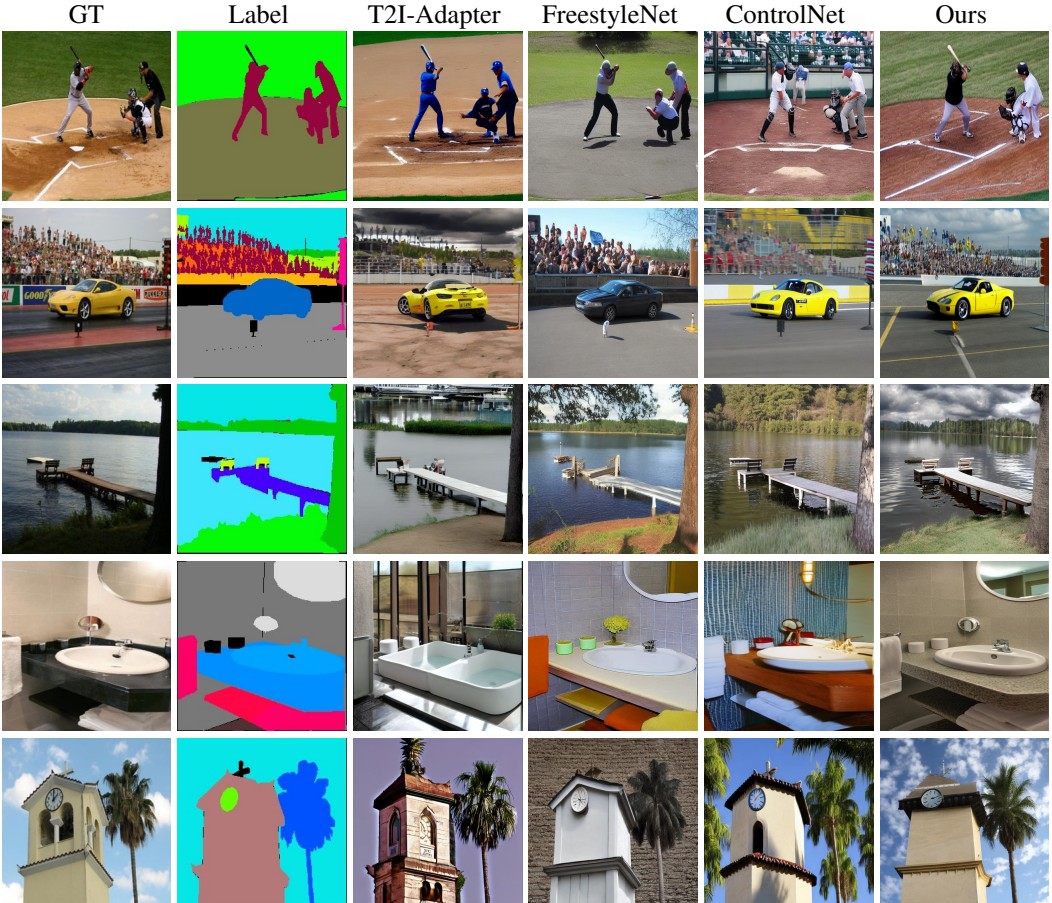

Figure 8: Qualitative comparison of faithfulness to the layout condition between different L2I methods on ADE20K. Our ALDM can comply with the label map consistently, while the other may ignore the ground truth label map and hallucinate, e.g., synthesizing trees in the background (see the third row).

## C.2 IMPROVED DOMAIN GENERALIZATION

More qualitative visualization on improved domain generalization is shown in Fig. 16. By employing synthetic data augmentation empowered by our ALDM, the segmentation model can make more reliable predictions, which is crucial for real-world deployment.

## D FAILURE CASES

As shown in Fig. 17, when editing the attribute of one object, it could affect the other objects as well. Such attribute leakage is presumably inherited from Stable Diffusion, which has been observed in prior work (Li et al., 2023a) with SD as well. Using a larger UNet backbone e.g. SDXL (Podell et al., 2023) or combining with other techniques, e.g., inference time latent optimization (Chefer et al., 2023; Li et al., 2023a) may mitigate this issue. This is an interesting open issue, and we would leave this for future investigation. Depsite the improvement on layout aligment, ALDM is not yet perfect and may not seamlessly align with given label map, especially when the text prompt is heaivly edited to a rare scenario.

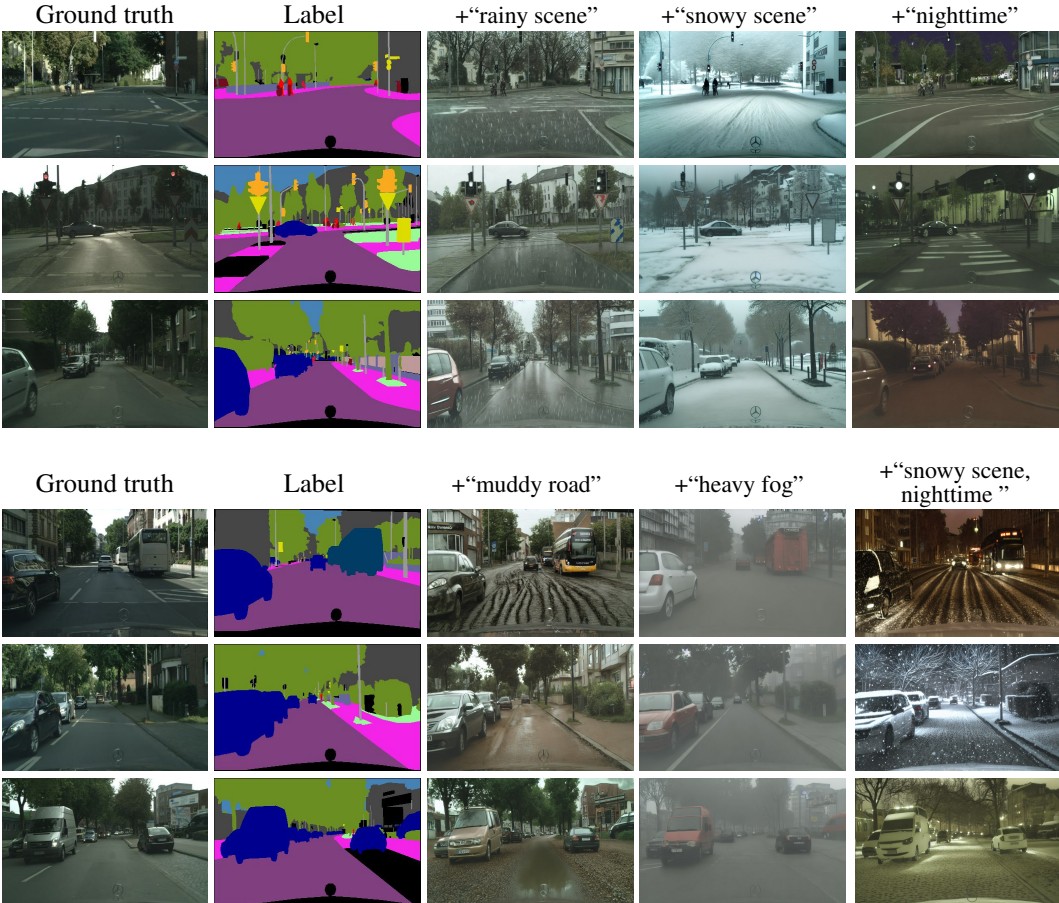

Figure 9: Visual examples of Cityscapes, synthesized by ALDM via various textual descriptions, which can be further utilized on downstream tasks.

# E    DISCUSSION & FUTURE WORK

## E.1    THEORETICAL DISCUSSION

In the proposed adversarial training, the denoising UNet of the diffusion model can be viewed as the generator, the segmentation model acts as the discriminator. For the diffusion model, the discriminator loss is combined with the original reconstruction loss, to further explicitly incorporate the label map condition. Prior works (Gur et al., 2020; Larsen et al., 2016; Xian et al., 2019) have combined VAE and GAN, and hypothesized that they can learn complementary information. Since both VAE and diffusion models (DMs) are latent variable models, the combined optimization of diffusion models with an adversarial model follows this same intuition - yet with all the advantages of DMs over VAE. The combination of the latent diffusion model with the discriminator is thus, in principle, a combination of a latent variable generative model with adversarial training. In this work, we have specified the adversarial loss such that relates our model to optimizing the expectation over $x_0$ in Eq. (6), and for the diffusion model, we refer to the MSE loss defined on the estimated noise in Eq. (2), which can be related to optimizing $x_0$ with respect to the approximate posterior $q$ by optimizing the variational lower bound on the log-likelihood as originally shown in DDPM (Ho et al., 2020). Our resulting combination of loss terms in Eq. (7) can thus be understood as to optimize over the weighted sum of expectations on $x_0$.

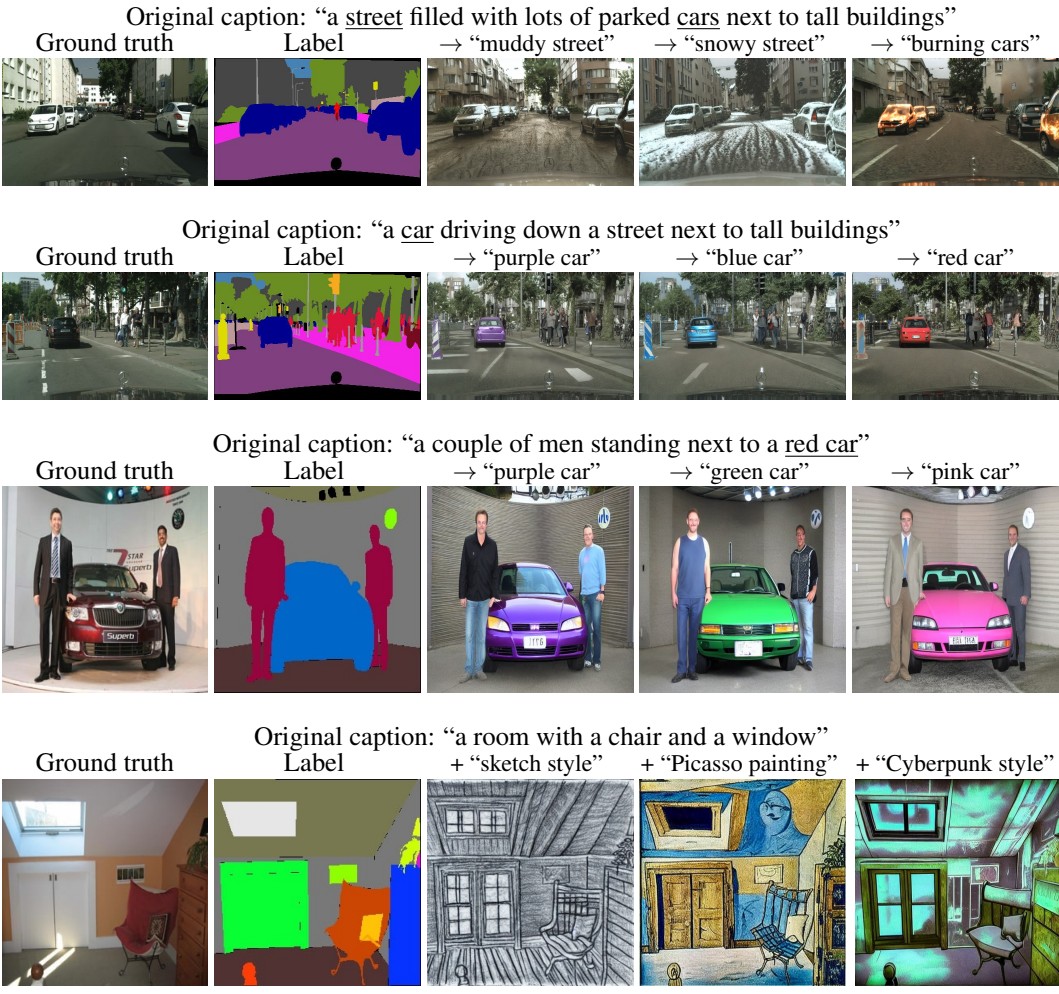

Figure 10: Visual examples of text controllability with our ALDM. Based on the original image captions generated by BLIP model, we can directly modify the underlined objects (indicated as →), or append a postfix to the caption (indicated as +). Our ALDM can accomplish both local attribute editing (e.g., car color) and global image style modification (e.g., sketch style).

## E.2 FUTURE WORK

In this work, we empirically demonstrated the effectiveness of proposed adversarial supervision and multistep unrolling. In the future, it is an interesting direction to further investigate how to better incorporate the adversarial supervision signal into diffusion models with thorough theoretical justification.

For the multistep unrolling strategy, we provided a fresh perspective and a crucial link to the advanced control algorithm - MPC, in Sec. 3.2. Witnessing the increasing interest in Reinforcement learning from Human Feedback (RLHF) for improving T2I diffusion models (Fan et al., 2023; Xu et al., 2023), it is a promising direction to combine our unrolling strategy with RL algorithm, where MPC has been married with RL in the context of control theory (Wang et al., 2023a) to combine the best of both world. In addition, varying the supervision signal rather than adversarial supervision, e.g., from human feedback, powerful pretrained models, can be incorporated for different purposes and tailored for various downstream applications. As formulated in Eq. (10), we simply average the losses at different unrolled steps, similar to simplified diffusion MSE loss (Ho et al., 2020). Future development on the time-dependent weighting of losses at different steps might further boost the effectiveness of the proposed unrolling strategy.

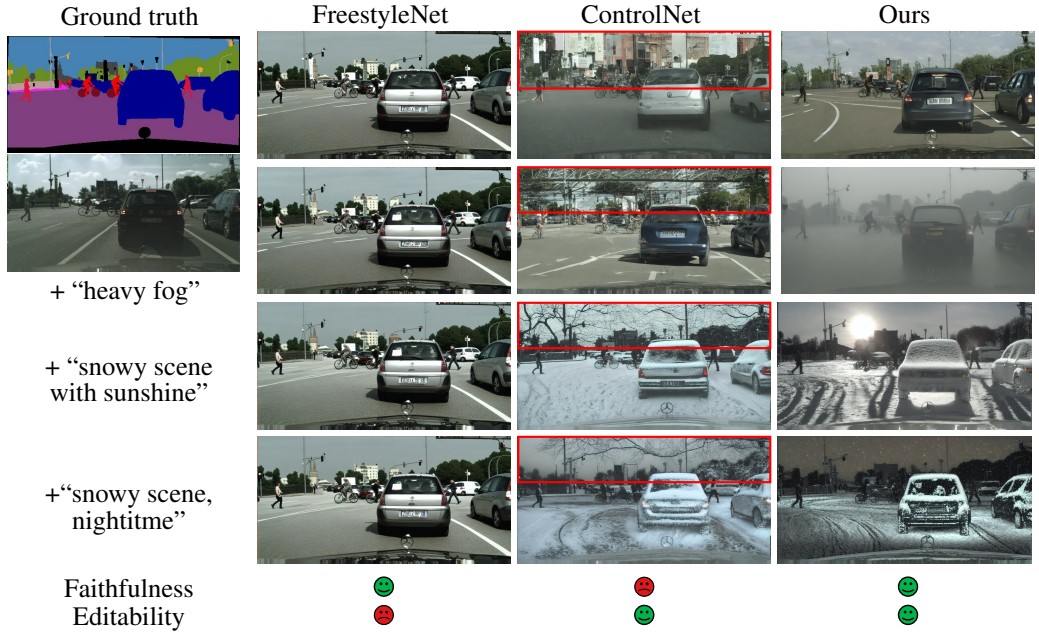

Figure 11: Qualitative comparison of text editability between different L2I diffusion models on Cityscapes. FreestyleNet exhibits little variability via text control. ControlNet often does not adhere to the layout condition, e.g., synthesizing buildings (1st row) or trees (2nd and 3rd row) where the ground truth label map is sky. In contrast, Our ALDM can synthesize samples well complied with both layout and text prompt condition.

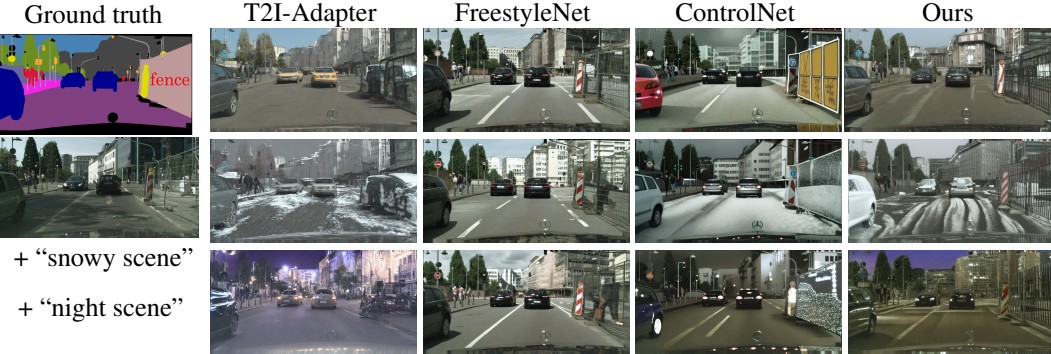

Figure 12: Qualitative comparison of text editability between different L2I diffusion models on Cityscapes.

Last but not the least, with the recent rapid development of powerful pretrained segmentation models such as SAM (Kirillov et al., 2023), autolabelling large datasets e.g., LAION-5B (Schuhmann et al., 2022) and subsequently training a foundation model jointly for the T2I and L2I tasks may become a compelling option, which can potentially elevate the performance of both tasks to unprecedented levels.

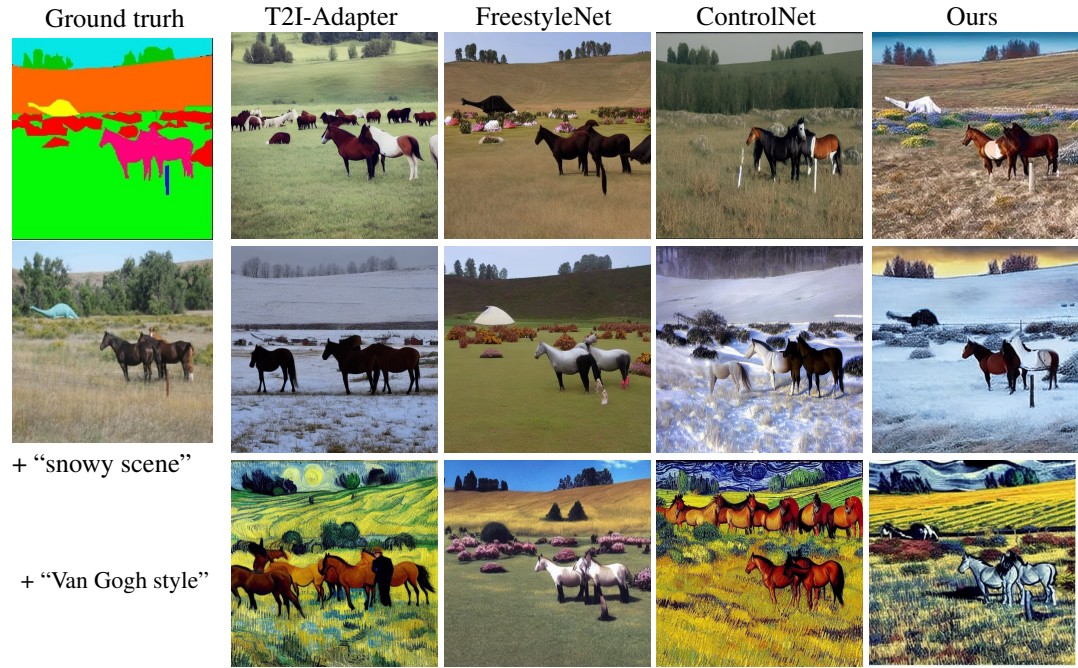

Figure 13: Qualitative comparison of text editability between different L2I diffusion models on ADE20K. ALDM can synthesize samples well complied with the layout and text prompt.

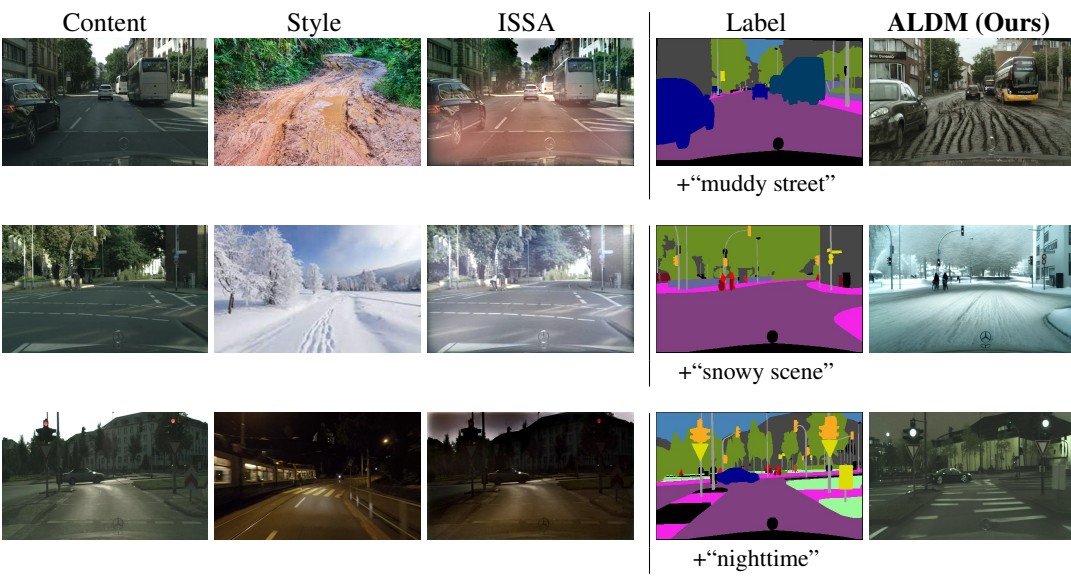

Figure 14: Comparison between our ALDM and GAN-based style-transfer method ISSA (Li et al., 2023b). It can be seen that ALDM can produce more realistic results with faithful local details, given the label map and text. In contrast, style transfer methods require two images, and mix them on the global color style, while the local details, e.g., mud, and snow may not be faithfully transferred.

| Ground truth | Label | Predicted $\hat{x}_0^{(t)}$ | Seg. Pred. |
|---|---|---|---|

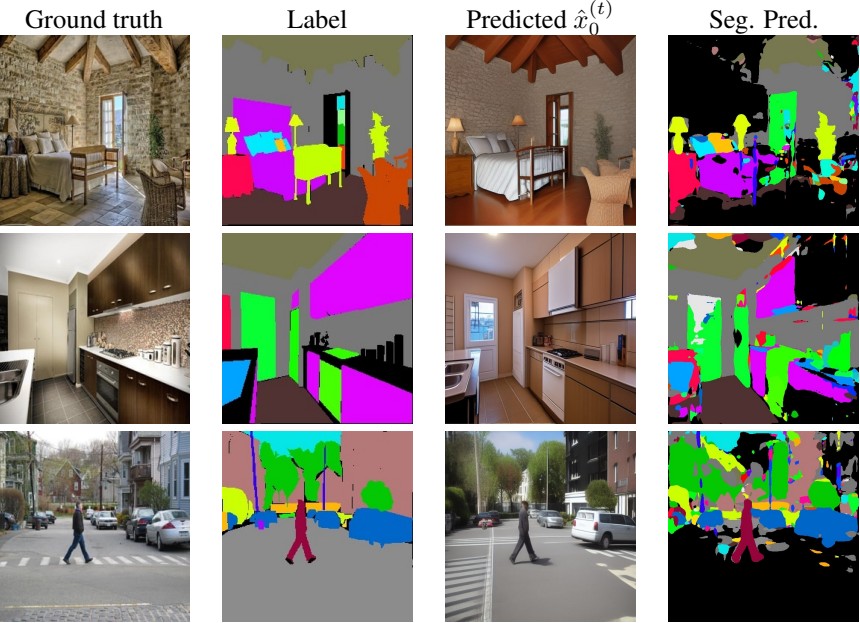

Figure 15: Visualization of discriminator predictions on the estimated clean image $\hat{x}_0^{(t)}$ at $t = 5$. Black in the ground truth label map represents unlabelled piels, while in the last segmentation predicion column black indicates the fake class predicitons.

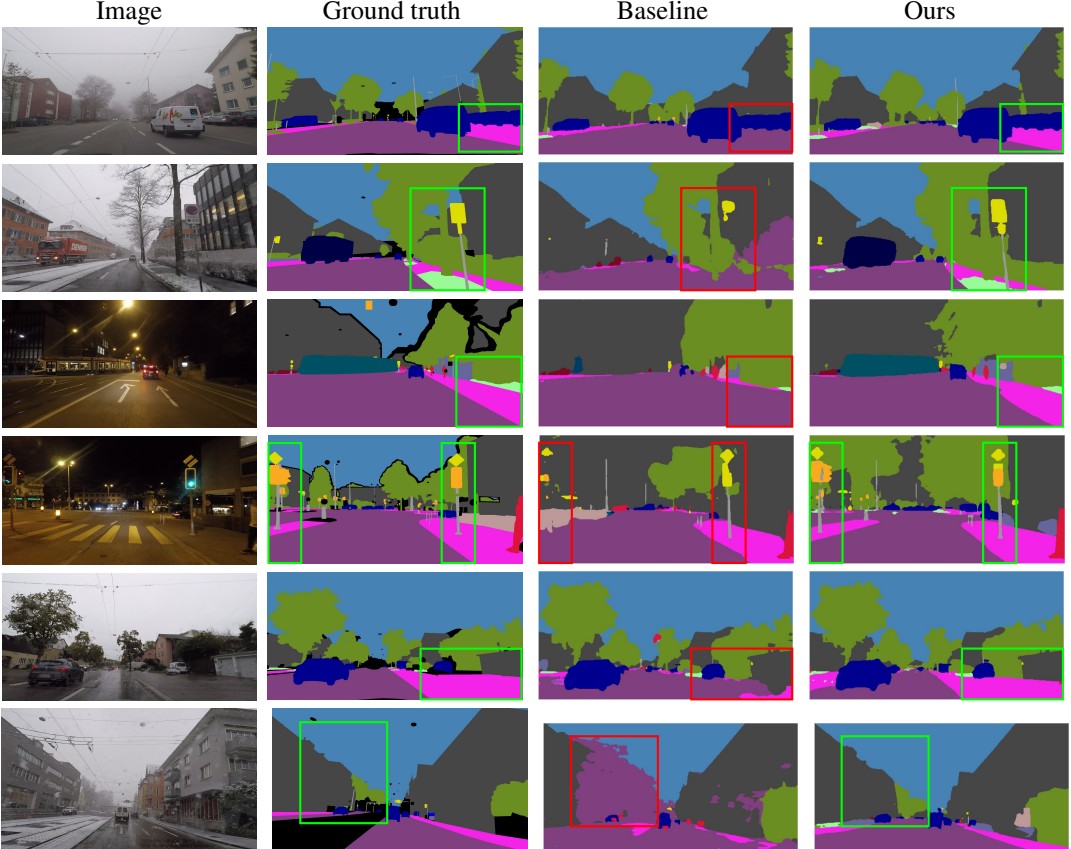

Figure 16: Semantic segmentation results of Cityscapes → ACDC generalization using HRNet. The HRNet is trained on Cityscapes only. Augmented with samples synthesized by ALDM, the segmentation model can make more reliable predictions under diverse unseen conditions, which is crucial for deployment in the open-world.

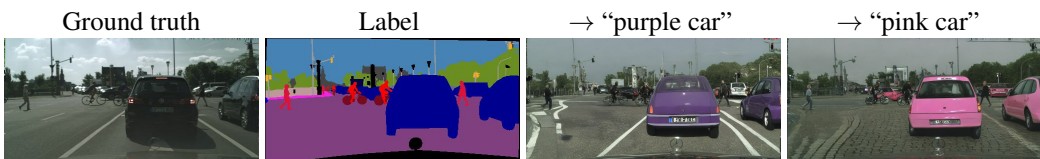

Figure 17: Editing failure cases. When editing the attribute of one object, it may leak to other objects as well. For instance, the color of the car on the right is modified as well.

