# OpenReview forum: "Adversarial Supervision Makes Layout-to-Image Diffusion Models Thrive"
_ICLR.cc/2024/Conference — ICLR 2024 poster_

### Official Review · Reviewer_VHY8 · 2023-10-23

**Soundness:** 3 good
**Presentation:** 1 poor
**Contribution:** 3 good
**Rating:** 6
**Confidence:** 4

**Summary:**

This paper aims for a better layout-to-image method. It applies a segmentation-based discriminator (Sushko et al., 2022) to the diffusion generator on the pixel-level alignment between the denoised image and the input layout. In addition, it proposes multistep unrolling, predicting the clean image at multiple timesteps and apply the segmenter-based discriminator. The experiments are on the classical segmentation datasets ADE20K and Cityscapes. The model exhibits comparable pixel level alignment and image fidelity. Different type of L2I synthesis adaptation models and segmenters are tested, to demonstrate its effectiveness.

**Strengths:**

1. The proposed method seems straightforward and effective.
2. The paper contains thorough ablation tests on different settings (different L2I models and segmenters) and different hyperparameters.

**Weaknesses:**

1. As mentioned in the Failure Cases, when editing the attribute of one object, it could affect the other objects as well. It is claimed to be inherited from Stable Diffusion.
2. Despite thorough ablation tests, the paper does not give any insights on the experiments. The text in the experiments only describes the results instead of analyzing the phenomenon. Thus, I think it shows limited contribution to the community. I suggest to delete the plain description of the results, as we can all see from the tables and figure captions, but add more analysis and insights of why it can work.

**Questions:**

1. When using a frozen segmenter, mIoU is higher and FID is also higher. Why does it happen, any insight?
2. It is mentioned in the limitations that editing one object may also affect others. However, in Figure 4, it shows better local controllability than ControlNet and FreestyleNet. Any insights why? Because of better pixel alignment? If that's the case, is better mIoU always means better local controllability? For example, can the one trained on frozen segmenter exhibits better local controllability?
3. Why larger number of unrolling steps is always better? Have you tried any K>9? Do all steps contribute the similar gradient magnitude or some of them is more important? What if we omit some of the earlier steps?

---

> ### Author Response · Authors · 2023-11-16
> **Response to Reviewer VHY8 (1/2)**
>
> We genuinely thank the reviewer for positively assessing our work and providing constructive feedback. We would like to provide detailed answers in the following to further clarify the remaining concerns:
>
> > (W.1) As mentioned in the Failure Cases, when editing the attribute of one object, it could affect the other objects as well. It is claimed to be inherited from Stable Diffusion.
>
> We believe that the attribute/concept binding issue, i.e., editing the attribute of one object could affect the other objects as well, is an inherited problem from Stable Diffusion (SD), as also been reported in prior SD based text-to-image works [1,2,3,4]. Using words to describe detailed changes at specific locations can be lengthy and inaccurate, therefore realizing such edit via text can be difficult. SD still cannot accomplish the task perfectly, either ignoring the text or mapping the text description to different objects. With the further scaling of the denoising UNet, e.g., SDXL [5], such issue might be mitigated but still exists as mentioned in their limitation section. One hypothesis for the binding issue lies in the pretrained text encoders, which tend to compress information into a fixed number of tokens. Employing inference time optimization [2,3] could resolve the problem to a certain extent. Nevertheless, we believe this is an interesting yet open research question.
>
> [1] Feng, Weixi, et al. "Training-Free Structured Diffusion Guidance for Compositional Text-to-Image Synthesis." ICLR. 2022.
>
> [2] Chefer, Hila, et al. "Attend-and-excite: Attention-based semantic guidance for text-to-image diffusion models." ACM Transactions on Graphics (TOG) 42.4 (2023): 1-10.
>
> [3] Li, Yumeng, et al. "Divide \& bind your attention for improved generative semantic nursing." arXiv preprint arXiv:2307.10864 (2023).
>
> [4] Rassin, Royi, et al. "Linguistic Binding in Diffusion Models: Enhancing Attribute Correspondence through Attention Map Alignment." arXiv preprint arXiv:2306.08877 (2023).
>
> [5] Podell, Dustin, et al. "Sdxl: Improving latent diffusion models for high-resolution image synthesis." arXiv preprint arXiv:2307.01952 (2023).
>
> > (W.2) Despite thorough ablation tests, the paper does not give any insights on the experiments. The text in the experiments only describes the results instead of analyzing the phenomenon. Suggestion to add more analysis and insights of why it can work.
>
> We thank the reviewer for pointing this out to help us further improve the manuscript quality. We will reiterate some of the insights presented in the Introduction and Method sections, and connect it better with the experiments. To provide a short summary of the important insights: ALDM improves upon the other methods from the training pipeline: the adversarial supervision explicitly leverages the label map condition; and the unrolling strategy can consider more sampling steps during the learning process, bridging the gap between training and the inference time sampling. The combination of both enables the model to well comply with the condition consistently over a time window, leading to the improvement on the alignment in the final result.
>
> > (Q.1) When using a frozen segmenter, mIoU is higher and FID is also higher. Why does it happen, any insight?
>
> When using a frozen segmenter, the diversity of the generated images is very limited, as shown in Fig 6. in the Appendix. The generator (i.e., the diffusing UNet) tends to learn a mean mode to cheat the discriminator (a frozen segmenter in this case), leading to the mode collapse issue. Such mean class mode is essentially easier for the segmenter to classify, yielding high mIoU. Since the FID metric assesses both visual quality and diversity,  such phenomenon is reflected in the high FID score. We have clarified this better in the revision as well.

---

> > ### Author Response · Authors · 2023-11-16
> > **Response to Reviewer VHY8 (2/2)**
> >
> > > (Q.2) It is mentioned in the limitations that editing one object may also affect others. However, in Figure 4, it shows better local controllability than ControlNet and FreestyleNet. Any insights why? Because of better pixel alignment? If that's the case, is better mIoU always means better local controllability? For example, can the one trained on frozen segmenter exhibits better local controllability?
> >
> > Our ALDM can improve the alignment between the synthesized image and the label map, which is beneficial for achieving more accurate local editing on a specific object area. Yet, the commonly observed binding issue from Stable Diffusion [1,2,3,4], i.e., editing one object may affect another object simultaneously, is still inherited. Despite ALDM improves on the pixel alignment due to the adversarial supervision and unrolling, the inherited binding issue cannot be eliminated completely, as shown in the limitation (Fig. 17). It can be seen that although the cars have been properly synthesized within the correct label map regions, extra editing such as color changes via text cannot be accurately enforced to the specific car. Improving alignment with the label map alone is not sufficient to improve the binding. One may seek to link the text and location in the label map or incorporate additional instructions. When using a pretrained frozen segmenter, the generator learns a mean class mode and heavily loses the diversity, which influences its text controllability. Our intuition is aligned with the quantitative results: we measured the text editability via the TIFA score when using a frozen segmenter, where it only achieves TIFA=0.556, which is significantly lower than ALDM with TIFA=0.888 as presented in Table 2.
> >
> >
> > > (Q.3) Why larger number of unrolling steps is always better? Have you tried any K $>$ 9?
> >
> > Larger unrolling step K can involve more sampling steps in the learning objective, better imitating the inference time sampling and encouraging more consistent adherence to the label map condition. Due to the linear increase of the unrolling overhead, we limit the number of steps to $K=9$ in our experiments. Nonetheless, post submission, we also experimented with more unrolling steps on ADE20K, using $K=15$, where mIoU was further improved (see Table 5 in Appendix, and listed below).
> >
> > | ControlNet | K=0 | K=6 |K=9 |K=15 |
> > |--|--|--|--|--|
> > | 30.4 |34.0|34.5|36.0|36.9|
> >
> > > (Q.4) Do all unrolling steps contribute the similar gradient magnitude, or some of them is more important? What if we omit some of the earlier steps?
> >
> > We simply average the discriminator loss across different unrolling steps, i.e., equal contribution, as formulated in Eq. (10), similar to simplified diffusion MSE loss proposed in DDPM [6]. Nevertheless, it is an interesting future direction to develop more advanced time-dependent weighting to further improve the algorithm. We thank the reviewer for the inspiring feedback, and we have included this in our newly added "Future Work" section (see Appendix E).
> > During training, the initial step $t$ before the unrolling is thus randomly sampled, since early denoising steps are also important for determining the content of the image, as observed in prior work [7].
> >
> > [6] Ho, Jonathan, Ajay Jain, and Pieter Abbeel. "Denoising diffusion probabilistic models." NeurIPS (2020): 6840-6851.
> >
> > [7] Hertz, Amir, et al. "Prompt-to-Prompt Image Editing with Cross-Attention Control." ICLR. 2022.

---

> > > ### Author Response · Authors · 2023-11-21
> > > **Thank you for the review & A kind reminder**
> > >
> > > Dear Reviewer VHY8,
> > >
> > > We sincerely appreciate the precious review time and valuable comments. This is just a friendly reminder that the discussion period is nearing its end.
> > >
> > > We would greatly appreciate it if you could confirm whether our responses have adequately addressed your concerns and consider the possibility of revising the score accordingly.
> > >
> > > Please let us know if you have any further questions. We remain committed to engaging in discussions and continually improving our submission.

---

> > > > ### Comment · Area_Chair_qk38 · 2023-11-21
> > > > **Dear reviewer**
> > > >
> > > > Please try your best to engage during the discussion period

---

> > > ### Comment · Reviewer_VHY8 · 2023-11-22
> > > **Official comment for the authors**
> > >
> > > Thanks for the additional experiments and the clarification. Most of my concerns have been addressed (except the analysis of different steps, and writing). Therefore I keep my original rating for weak acceptance, but increase the contribution score.

---

### Official Review · Reviewer_rjfS · 2023-10-30

**Soundness:** 3 good
**Presentation:** 3 good
**Contribution:** 3 good
**Rating:** 6
**Confidence:** 4

**Summary:**

The paper "Adversarial Supervision Makes Layout-to-Image Diffusion Models Thrive" proposes to augment diffusion models with adversarial methods to achieve a model that is controlable from text and segmentation map layout simultaneously. In particular, the authors propose two additions to fine-tune a Stable Diffusion model: 1) a recent adversarial learning methods incorporating segmentation networks is applied to the outputs of the diffusion network, and 2) the diffusion process is unrolled, and the segmentation network is applied already to intermediate de-noised steps. It is shown in the paper that there's incremental value in both. The method is evaluated on two datasets (ADE20K and Cityscapes), and the proposed method is showns to be superior to several recent baselines (T2I-Adapter, FreestyleNet and ControlNet) qualitatively, as well as quantitatively. Finally, the method is applied on a domain generalization task, and is shown to perform best against the baselines there.

**Strengths:**

+ the paper is well written, intuitive motivations are given, and it is well understandable.
+ the proposed method is shown to work well. It shows both qualititatively better alignment and quantitative improvements.
+ it also shows very good results on domain generalization.

**Weaknesses:**

- it's unclear if careful tuning of a frozen segmentation network's impact on the total loss wouldn't be competitive to the proposed adversarial approach. I.e. in Table 3, the frozen UperNet achieves the best mIoU, but much worse FID - consistent with the hypothesis that the impact of the segmentation network is just too strong. Verying the impact of the segmentation loss relative to the diffusion loss while fine-tuning would clarify this.
- The introduction gave a good intuition for the whole method. However, the combination of diffusion models with adversarial methods remains ad-hoc, and there's no theoretical justification for the validaty of the approach given. It should be either worked out, or at least added for future work that propoer understanding of diffusion-adversarial coupling should be investigated (to make these methods work best together).

Smaller things:
- Figure 1 lowest right image: this example shows that the method doesn't yet work perfectly. The top part of the truck is a building (see the windows and the 'roof structure'). It should be pointed out in the paper that some failed cases still persist (even if harder to see).
- In "Related Work" you write "more attention has been devoted to leveraging pretrained knowledge for the L2I task and using diffusion models"; I think this is an important point, and should already be part of the motivation of the general method in the Introduction.

**Questions:**

- why don't you train a diffusion model from scratch for the task of L2I and T2I simultaneously? Recent segmentation models are very powerful, and can produce the segmentation maps needed to augment the dataset (e.g. LAION-5B). I'd expect that to work best.
- for the case of a frozen pre-trained segmenter: did you train a diffusion model plus an additional loss for the segmentation model? That is surprising, because I wouldn't have expected the diffusion model to collapse in such case (the same training method/loss is still there).

---

> ### Author Response · Authors · 2023-11-16
> **Response to Reviewer rjfS (1/2)**
>
> We sincerely thank the reviewer for the overall positive assessment and provided insightful feedback. We have revised the manuscript based on your constructive comments. In what follows, we provide our response to individual questions.
>
> > (Q. 1) Train a diffusion model from scratch for the task of L2I and T2I simultaneously. Recent segmentation models can produce the segmentation maps needed to augment the dataset (e.g. LAION-5B).
>
> We agree that training a diffusion model for L2I and T2I jointly is a promising idea, and the recent segmentation models such as SAM have demonstrated impressive zero-shot performance, and in many cases can provide reliable label maps for training. Unfortunately, in our lab we have very limited computational resources, thus large-scale training, e.g., 256 A100 GPUs with 150000 hours reported for Stable Diffusion, is far beyond our computing capability. In contrast, fine-tuning only requires 2 A100 GPUs, which is affordable for us. As not having enough resources to train Stable Diffusion from scratch is not an uncommon status for many labs in the world, developing fine-tuning techniques is valuable as well. Nevertheless, we thank the reviewer's valuable suggestion, and we have incorporated this into the newly added "Future Work" discussion in Appendix E.
>
> > (Q. 2 & W. 1) (W.1) It's unclear if careful tuning of a frozen segmentation network's impact on the total loss wouldn't be competitive to the proposed adversarial approach.
> (Q. 2) for the case of a frozen pre-trained segmenter: did you train a diffusion model plus an additional loss for the segmentation model?
>
> Our initial assumption was the same, however, we couldn't manage to resolve the issue after tuning the hyperparameters. As shown below, when decreasing the weighting $\lambda$ of the frozen segmenter loss, the mIoU drops heavily while the FID has minor improvement. The training loss is the same as formulated in Eq. (7), which is a combination of diffusion MSE loss and the segmenter loss. The only difference is for our proposed ALDM the segmenter as a discriminator is also updated with the diffusion model instead of being frozen. In the adversarial game, it is important that the discriminator and generator strike a balance to continuously improve each other. While using a frozen segmenter, the discriminator loses the chance to update itself, and the powerful generator (Stable Diffusion UNet) can freely find a cheating path, leading to the mode collapse issue, as observed in GANs [1]. The mean mode yields limited diversity thus high FID, and it's easier for the segmenter to classify the class mean mode thus the high mIoU.
>
> | Method| mIoU | FID|
> |--|--|--|
> | $$\lambda=1e-2$$ |50.8|40.2|
> | $$\lambda=1e-4$$ |39.2|39.8|
> | ALDM (Ours)|36.0|30.2|
>
> [1] Thanh-Tung, Hoang, and Truyen Tran. "Catastrophic forgetting and mode collapse in GANs." 2020 International Joint Conference on Neural Networks (IJCNN). IEEE, 2020.

---

> > ### Author Response · Authors · 2023-11-16
> > **Response to Reviewer rjfS (2/2)**
> >
> > > (W. 2) The introduction gave a good intuition for the whole method. However, the combination of diffusion models with adversarial methods remains ad-hoc, and there's no theoretical justification for the validity of the approach given. It should be either worked out, or at least added for future work that proper understanding of diffusion-adversarial coupling should be investigated (to make these methods work best together).
> >
> > We thank the reviewer for acknowledging that the introduction provides a good intuition, and for providing the constructive suggestion. We acknowledge the value of having a carefully derived mathematical framework for the problem. In the proposed adversarial training, the denoising UNet of the diffusion models can be viewed as the generator, the segmentation based model acts as the discriminator. For the diffusion model, the discriminator loss is combined with the original reconstruction loss, to further explicitly incorporate the label map condition. Prior works [2,3] have combined VAE and GAN, and hypothesized that they can learn complementary information. Since both VAE and Diffusion Models (DMs) are latent variable models, the combined optimization of DMs with an adversarial model follows this same intuition - yet with all the advantages of DMs over VAE.
> >
> > The combination of the latent diffusion model with the discriminator is thus, in principle, a combination of a latent variable generative model with adversarial training. In our submission, we  specified the adversarial loss such that relates our model to optimizing the expectation over ${x}_0$ in Equation (6), while for the diffusion model, we refer to the MSE loss defined on the estimated noise. In the original DDPM paper [4], it has been shown how the MSE loss in our Equation (2) relates to optimizing ${x}_0$ with respect to the approximate posterior $q$ by optimizing the variational lower bound on the log-likelihood (refer to their equations (1)-(14), where our Equation (2) corresponds to their Equation (14)). For the relationship of the MSE loss to the approximate posterior optimization, we can in particular refer to Equation (5) in DDPM [4]. Our resulting combination of loss terms in Equation (7) can thus be understood as to optimize over the weighted sum of expectations on ${x}_0$ given in our Equation (6) and Equation (5) in DDPM [4].
> >
> > We thank the reviewer for pointing this out and help us to improve the manuscript quality. We rewrote our Equation (2) such that its relationship to optimizing an expectation over $x_0$ becomes clearer, and we have included the above discussion of the theoretical framework in the new section "Discussion \& Future Work" (see Appendix E). May we know if this addresses the concern?
> >
> > [2] Larsen, Anders Boesen Lindbo, et al. "Autoencoding beyond pixels using a learned similarity metric." International conference on machine learning. PMLR, 2016.
> >
> > [3] Xian, Yongqin, et al. "f-vaegan-d2: A feature generating framework for any-shot learning." Proceedings of the IEEE/CVF conference on computer vision and pattern recognition. 2019.
> >
> > [4] Ho, Jonathan, Ajay Jain, and Pieter Abbeel. "Denoising diffusion probabilistic models." Advances in neural information processing systems 33 (2020): 6840-6851.
> >
> > > Smaller things
> >
> > We are grateful for the reviewer's valuable suggestions and have incorporated the modifications accordingly to improve the manuscript. As mentioned in (W.4), we agree that "more attention has been devoted to leveraging pretrained knowledge for the L2I task and using diffusion models" is an important point, and have highlighted the relevant part in the "Introduction" in blue now. To address (W.3), we additionally point out that ALDM is not yet perfect in the failure cases section (see Appendix D).

---

> > > ### Author Response · Authors · 2023-11-21
> > > **Thank you for the review & A kind reminder**
> > >
> > > Dear Reviewer rjfS,
> > >
> > > We sincerely appreciate your constructive comments for helping us improve our submission. This is just a gentle reminder as the discussion deadline is approaching.
> > >
> > > We have incorporated your valuable suggestions into our revision. If you find that our responses have adequately addressed your concerns, we would greatly appreciate your consideration of a potential score revision. Please let us know if there are any additional aspects we could further improve.

---

> > > > ### Comment · Area_Chair_qk38 · 2023-11-21
> > > > **Dear reviewer**
> > > >
> > > > Please try your best to engage during the discussion period

---

> > > ### Comment · Reviewer_rjfS · 2023-11-21
> > >
> > > Thanks for the elaborate discussion on the frozen segmentation network, the discussion on the interaction of diffusion and adversarial methods, and adding Appendix E.
> > > This settles most of my questions. I'll update my final score after discussion with the other reviewers.

---

### Official Review · Reviewer_d5qA · 2023-10-30

**Soundness:** 3 good
**Presentation:** 3 good
**Contribution:** 3 good
**Rating:** 6
**Confidence:** 4

**Summary:**

This paper proposes to adopt a ControlNet architecture for better semantic image synthesis using an adversarial discriminator (as in OASIS) on the per-pixel label maps. Furthermore, a multistep unrolling mechanism is presented so that adversarial supervision takes into account several denoising steps to improve the signal at low noise levels.

**Strengths:**

- This paper is well-structured and well-written.
- Ideas and results are presented clearly.
- Adapting a pre-trained diffusion model for semantic image synthesis is interesting and challenging.

**Weaknesses:**

There are several issues with the presented work.

- The technical contributions of this manuscript are limited. Adversarial supervision on the semantic maps is identical as in OASIS. Multistep unrolling is computationally very expensive (scales linearly and hence can take up x9 longer), and has a small effect on the performance.
- The results (while improving over the diffusion baselines) are behind OASIS, a GAN from 2020.
- The motivation to use a strong text-to-image model and adapt it for semantic image synthesis is flawed. The paper motivates it by enabling text-conditioned content and style transfer, but the semantic mask completely specifies the content. Hence, the application is reduced to text-guided style and color transfer.
- Furthermore, the proposed model does not perform style transfer well. When changing to a snowy scene, the whole image and all objects are resampled. Local editing is also not possible. Instead, the whole image is affected when changing "a red van" to "a burning van".
- A simple baseline combining OASIS and a state-of-the-art style transfer model should be considered.

Side remarks:
- Layout-to-image is usually referred to as a task that transforms a list of bounding boxes and class labels into an image [1,2]. This paper tackles semantic image synthesis where the input is a label mask (each pixel is labelled).
- The paper states that Stable Diffusion based models do not comply well with layout input, but see [3,4]

[1] Image Synthesis From Reconfigurable Layout and Style, https://arxiv.org/abs/1908.07500
[2] LayoutDiffusion: Controllable Diffusion Model for Layout-to-image Generation, https://arxiv.org/abs/2303.17189
[3] SpaText: Spatio-Textual Representation for Controllable Image Generation, https://arxiv.org/abs/2211.14305
[4] ReCo: Region-Controlled Text-to-Image Generation, https://arxiv.org/abs/2211.15518

**Questions:**

-

---

> ### Author Response · Authors · 2023-11-16
> **Response to Reviewer d5qA  (1/2)**
>
> We sincerely thank Reviewer d5qa's acknowledging the paper is well-written, the topic is interesting and challenging. We believe there is a misunderstanding on the motivation of our approach, and thus would like to reiterate the motivation, our technical contributions and placement of our approach in the scope of the semantic image synthesis work, both diffusion-based and GAN-based. We believe we point out an important issue in the current diffusion model pipeline and the proposed training strategy has proven its effectiveness on different diffusion model based approaches.  We next would like to address individual concerns:
>
> > (1)  The technical contributions of this manuscript are limited.
>
> To the best of our knowledge, our adversarial supervision with the multistep unrolling strategy has not been proposed before for improving layout-to-image diffusion models. Our approach is indeed inspired by OASIS, which we have adequately acknowledged in the paper, i.e., cited it in multiple sections (introduction, related work, method, etc.) in the main paper.  However, incorporating the broad concept of discriminator loss into a completely different type of generative model, i.e., diffusion model, is by no means trivial.
>
> The traditional diffusion model suffers from poor alignment due to the suboptimal training objective without explicit supervision based on the label map, and its sampling characteristics also make consistent adherence to the conditional layout with time difficult. Little attention has been paid to improve the original diffusion reconstruction loss. Therefore, the proposed adversarial supervision and multistep unrolling is a novel training design for diffusion models, to explicitly encourage consistent alignment to the label map over a time window, closing the gap with inference time sampling. **We believe our work focuses on the important yet underexplored aspects of diffusion models, i.e., the training strategy**, and we demonstrate the effectiveness of our proposal on different recent diffusion model-based methods, e.g., OFT (NeurIPS'23), T2I-Adapter (ArXiv'23-Mar) and SOTA ControlNet (ICCV'23 Best Paper) (see Table 1 and below). Especially, the multistep unrolling is  tailored for diffusion models, considering its sampling characteristic, which is significantly different from GANs.
>
> | Method| Cityscapes mIoU | ADE20K mIoU|
> |--|--|--|
> | OFT |48.9|24.1|
> | OFT + Ours |58.8|31.8|
> | T2I-Adapter|37.1|24.0|
> | T2I-Adapter + Ours|50.1|29.1|
> | ControlNet |55.2|30.4|
> | ControlNet + Ours |63.9|36.0|
>
> Regarding the computing overhead, as mentioned in the "Implementation details", we only apply the unrolling every 8 optimization steps, such that the additional overhead is manageable. It is important to mention that multistep unrolling is a training strategy that can largely improve the alignment with the label map, coming at no additional cost at inference time, which is more important for most applications.
>
> > (2) The results (while improving over the diffusion baselines) are behind OASIS, a GAN from 2020.
>
> Our goal is to improve text-to-image (T2I) diffusion models for semantic image synthesis.  We agree with the reviewer that this class of generative models underperforms GANs in terms of alignment, which we have shown in Table 8 in the Appendix. However, there are many benefits of using large-scale T2I diffusion models for this task, such as powerful text controllability. Our ALDM can naturally synthesize novel samples, which can be used for downstream applications such as improved domain generalization (see Table 4 and Fig. 5). In contrast, GAN alone can only synthesize samples resembling the trained domain. We believe our proposed training strategies can inspire the community and can be combined with advancing future diffusion model architecture designs, which are orthogonal to our work, leading to better results in the future.

---

> > ### Author Response · Authors · 2023-11-16
> > **Response to Reviewer d5qA (2/2)**
> >
> > > (3 & 5) Motivation to use a strong text-to-image model and adapt it for semantic image synthesis is flawed. The paper motivates it by enabling text-conditioned content and style transfer, but the semantic mask completely specifies the content. Hence, the application is reduced to text-guided style and color transfer. A simple baseline combining OASIS and a state-of-the-art style transfer model should be considered.
> >
> > Our ALDM is not only capable of global style changes but also local editing via text, as presented in Fig. 1, 4, 9-13. We have verified the effectiveness of synthesized novel samples on domain generalization task (in Table 4 and below), where we compare against GAN-based style transfer ISSA [1]. We thank the reviewer for the inspiring comment, we additionally compare ALDM against text-guided editing approach PnP-Diffusion [2] below as well. ALDM outperforms both methods and largely improves the generalization performance of the segmenters. A visual comparison between ALDM and the style transfer method has been added in the revision in Fig. 14, where it can be seen that ours can produce more realistic results with faithful local details, e.g., mud, snow. In contrast, the majority of the style transfer methods mix two images on the global color style, while the local details cannot be faithfully transferred (as also seen from Fig. 14).
> >
> > | Method| ACDC - HRNet | ACDC  - SegFormer|
> > |--|--|--|
> > | Baseline (Cityscapes) |41.48|47.04|
> > | ISSA (GAN-based style transfer) |50.05|52.45|
> > | PnP-Diffusion (Text-guided editing) |48.74|48.71|
> > | ALDM (Ours)|53.03|53.78|
> >
> > [1] Li, Yumeng, et al. "Intra-& extra-source exemplar-based style synthesis for improved domain generalization." International Journal of Computer Vision (2023): 1-20.
> >
> > [2] Tumanyan, Narek, et al. "Plug-and-play diffusion features for text-driven image-to-image translation." Proceedings of the IEEE/CVF Conference on Computer Vision and Pattern Recognition. 2023.
> >
> > > (4) Furthermore, the proposed model does not perform style transfer well. When changing to a snowy scene, the whole image and all objects are resampled. Local editing is also not possible. Instead, the whole image is affected when changing "a red van" to "a burning van".
> >
> > As we are tackling the task of semantic image synthesis rather than style transfer, it is not required or desired that object instances are preserved with varying text prompts, as long as the label map condition is fulfilled. ALDM is equipped with powerful text editability, while we also openly discussed limitations of the current approach and showed failure cases in Fig. 17.
> >
> >
> > > (side remarks 1) Layout-to-image is usually referred to as a task that transforms a list of bounding boxes and class labels into an image. This paper tackles semantic image synthesis where the input is a label mask (each pixel is labelled)
> >
> > We thank the reviewer for the references. We agree that the terminology is not well-defined, and we acknowledge that the semantic image synthesis is also correct [3,4]. Since we focused on diffusion models, as also clearly stated in the title, we have chosen to inherit the terminology from the recent FreestyleNet: Freestyle Layout-to-Image Synthesis (CVPR'23) which is a diffusion model conditioned on label maps. Nevertheless, we thank the reviewer for pointing this out, and we have now incorporated this in the related work in the revision.
> >
> > [3] Schönfeld, Edgar, et al. "You Only Need Adversarial Supervision for Semantic Image Synthesis." International Conference on Learning Representations. 2020.
> >
> > [4] Wang, Yi, et al. "Image synthesis via semantic composition." Proceedings of the IEEE/CVF International Conference on Computer Vision. 2021.
> >
> > > (side remarks 2) The paper states that Stable Diffusion based models do not comply well with layout input, but see ReCo and SpaText
> >
> > We thank the reviewer for the references. ReCo only tries to control some objects given a rough bounding box position, which does not require per-pixel alignment. As showcased in the limitation of SpaText already, it does not perform well for more complex cases. Meanwhile, we have conducted extensive evaluation against other recent SOTA methods, e.g., FreestyleNet (CVPR'23), OFT (NeurIPS'23), T2I-Adapter (ArXiv'23-Mar), ControlNet (ICCV'23 Best Paper), and improved upon it.
> >
> > We thank the reviewer for taking time on the review and rebuttal and providing thoughtful feedback. Please let us know if there are any remaining concerns we could address to help improve the score.

---

> > > ### Author Response · Authors · 2023-11-21
> > > **Thank you for the review & A kind reminder**
> > >
> > > Dear Reviewer d5qA,
> > >
> > > Thank you for your precious time in reviewing our paper and providing insightful comments. This is just a gentle reminder to revisit our responses, where we have diligently addressed your concerns. If you have any further questions, please do not hesitate to let us know.
> > >
> > > Your consideration of raising the score, in light of the clarification we've provided, would be highly appreciated.

---

> > > > ### Comment · Area_Chair_qk38 · 2023-11-21
> > > > **Dear reviewer**
> > > >
> > > > Please try your best to engage during the discussion period

---

> > > > > ### Author Response · Authors · 2023-11-22
> > > > > **Thanks Reviewer d5qA for reviewing & A final kind reminder**
> > > > >
> > > > > Dear Reviewer d5qA,
> > > > >
> > > > > As the discussion period enters its final day, we sincerely would like to receive the feedback from the reviewer and hope that there is still time to address any remaining concerns.
> > > > >
> > > > > Given the distinctiveness of your evaluations in comparison to those of the other reviewers, your feedback holds particular significance. If our responses have effectively addressed the concerns, your thoughtful reconsideration of adjusting the score accordingly would be greatly appreciated.
> > > > >
> > > > > If there are any additional questions, we are more than happy to address them and engage in further discussion with the reviewer in the remaining time.

---

> > > > > > ### Comment · Reviewer_d5qA · 2023-11-22
> > > > > > **Response**
> > > > > >
> > > > > > Dear authors, thank you for the detailed response, additional analysis, and patience. I have read the updated version, replies, and other reviews. My main concerns have been addressed thus I have decided to raise my score and suggest acceptance. The work presents a valuable contribution to the community in improving the controllability of existing large-scale t2i models.

---

> > > > > > > ### Author Response · Authors · 2023-11-22
> > > > > > > **Sincere Thanks to Reviewer d5qA**
> > > > > > >
> > > > > > > We genuinely appreciate the reviewer's reconsideration and recommendation to accept our paper. Your valuable feedback and engagement in the discussion have been of great significance to us. Thank you very much for your time and efforts in reviewing.

---

### Official Review · Reviewer_4k64 · 2023-10-31

**Soundness:** 3 good
**Presentation:** 3 good
**Contribution:** 3 good
**Rating:** 6
**Confidence:** 4

**Summary:**

This paper proposes to embed adversarial supervision into the training of a diffusion model conditioned on layout. By introducing adversarial supervision and the multi-step rolling strategy, the framework can get strong results, better than baseline methods like control net which does not explicitly use adversarial supervision. It also shows that the generated samples can boost the domain generalization for semantic segmentation tasks.

**Strengths:**

- The paper is well-written and easy to understand, the proposed method is simple but effective
- Adversarial supervision is not new though, it is the first time to be used in the context of diffusion model
- The experimental results are strong and are better than the baseline methods.

**Weaknesses:**

- The qualitative results seem to be much better than baseline methods, are they cherry-picked? A non-cherry picked results can better show that the proposed methods largely exceed the current baselines.
- What is the difference between Control-Net + Adv Supervision and multi-step rolling in Table 1 compared with ALDM?
- Can you show some outputs of the discriminator? Since it works in the latent space, we need to be more careful about what is happening inside.

**Questions:**

In the `weaknesses`

---

> ### Author Response · Authors · 2023-11-16
> **Response to Reviewer 4k64**
>
> We thank the reviewer for the overall positive assessment, especially acknowledging the simplicity and effectiveness of our method. In what follows, we address the individual concerns in detail:
>
> > (1) The qualitative results seem to be much better than baseline methods, are they cherry-picked?
>
> We selected representative examples to illustrate the problems in existing methods in the teaser figure, i.e., Fig 1. For other figures, we did not curate the results. We thank the reviewer to help us further improving the manuscript quality, and we have provided more comparisons in the Appendix. Besides, the quality comparison of random samples can also be seen in the extensive quantitative evaluation (e.g., Table 1), in which our method outperforms the other competitors.
>
> > (2) What is the difference between Control-Net + Adv Supervision and multi-step rolling in Table 1 compared with ALDM?
>
> There is no difference between Control-Net + Adv Supervision + multi-step rolling and ALDM in other tables. In general, ALDM (adversarial supervision plus multi-step unrolling) can be seen as an improved training pipeline for L2I diffusion models, which can be combined with different adaptation architecture designs as shown in Table 1. In other tables, we use ControlNet as the default model combined with adversarial supervision and multistep unrolling, to represent ALDM.  We thank the reviewer for pointing this out, and we have clarified accordingly in the revision (on Page 7 after Table 1).
>
> > (3) Can you show some outputs of the discriminator?
>
> We have added some discriminator predictions in Fig. 15. The discriminator takes the input of the decoder, which actually turns the latents into the pixel images, as mentioned in the "Implementation details". From Fig. 15 the discriminator can categorize some regions as "fake" class (in black), meanwhile, it is also fooled by some areas, where the discriminator produces reasonable segmentation predictions matched with ground truth labels. Therefore, the discriminator can provide useful feedback to the generator (diffusion model) that it should produce realistic results meanwhile complying with the given label map.

---

> > ### Comment · Reviewer_4k64 · 2023-11-16
> > **Thanks for the reply**
> >
> > The response answered most of my questions, I will keep and score and raise up my confidence. I will also further decide my final score by discussing it with other reviewers.

---

> ### Author Response · Authors · 2023-11-17
> **Thanks for the reviewer's prompt reply**
>
> We genuinely thank the reviewer for the prompt response. We are glad that our answers have improved your confidence that our work is above the acceptance bar.
>
> Please let us know if there are any new concerns we could address to further improve the score during the subsequent discussion. We remain committed to addressing any further questions to ensure the continued improvement of our manuscript.

---

### Author Response · Authors · 2023-11-22
**Thanks to all reviewers & A final kind reminder**

We genuinely thank all the reviewers for their time, effort, and dedication to the review process.

We are pleased to note that R1 (4k64) and R3 (rjfS) have kindly confirmed our rebuttal has effectively addressed their concerns, and they continue to support the acceptance of our paper. We greatly appreciate your positive feedback and your recognition of the improvements we have made in the revision based on your constructive comments.

As we approach the final day of the discussion phase, we again warmly welcome R2 (d5qA) and R4 (VHY8) to join this constructive dialogue. Your valuable feedbacks are highly appreciated, and we are committed to addressing any remaining concerns you may have.

---

### Author Response · Authors · 2023-11-23
**General Response - Thanks to all reviewers for their valuable feedback**

We sincerely thank all reviewers for their constructive feedback and comments. We are pleased to see that the reviewers found our method is straightforward, simple yet effective, works well both qualitatively and quantitatively (R1-4k64, R3-rjfS, R4-VHY8), the paper is well-written and intuitive motivations are given (R1, R2-d5qA, R3). We are equally glad that R4 has acknowledged our thorough ablation studies on different settings.

We are glad that after the discussion period, all reviewers have reached a consensus in recommending the acceptance of our paper and have acknowledged the value of our work to the community. Specifically, R1 has raised the confidence score towards acceptance; R2 has raised the overall score from 3 to 6; R3 has confirmed our elaborated responses and revisions have addressed most of the concerns; and R4 increased the contribution score.

We genuinely thank all the reviewers for their insightful suggestions, which have significantly improved the quality of our manuscript during the revision process. We also appreciate the AC's encouragement of the discussion.

---

### Meta-Review · Area_Chair_qk38 · 2023-12-05

**Metareview:**

The paper addresses the problem of improving alignment for layout-to-image models and suggests using a segmentation-based discriminator which provides feedback to the diffusion generator on alignment. The authors also suggest running the discriminator on several steps unrolled together. The authors then evaluate on ADE20K and Cityscapes and show improvements qualitatively and quantitatively. The results on Cityscapes seem strong -- the ADE20K results seem a bit weaker.

All reviewers suggest Acceptance and I don't see any reason to override them.

**Justification For Why Not Higher Score:**

The paper has somewhat limited technical contribution and the quantitative results could be stronger.

**Justification For Why Not Lower Score:**

The paper outperforms prior techniques and is a useful contribution to the community.

---

### Decision · Program_Chairs · 2024-01-16

Accept (poster)